# Structural basis for germline antibody recognition of HIV-1 immunogens

**Louise Scharf[1†], Anthony P West Jr[1], Stuart A Sievers[1‡], Courtney Chen[1], Siduo Jiang[1§], Han Gao[1], Matthew D Gray[2], Andrew T McGuire[2], Johannes F Scheid[3], Michel C Nussenzweig[3,4], Leonidas Stamatatos[2], Pamela J Bjorkman[1*]**

[1]Division of Biology and Biological Engineering, California Institute of Technology, Pasadena, United States; [2]Vaccine and Infectious Disease Division, Fred Hutchinson Cancer Research Center, Seattle, United States; [3]Laboratory of Molecular Immunology, The Rockefeller University, New York, United States; [4]Howard Hughes Medical Institute, The Rockefeller University, New York, United States

**\*For correspondence:** bjorkman@caltech.edu

**Present address:** [†]23andMe, Mountain View, United States; [‡]Kite Pharma, Santa Monica, United States; [§]D. E. Shaw Research, New York, United States

**Abstract** Efforts to elicit broadly neutralizing antibodies (bNAbs) against HIV-1 require understanding germline bNAb recognition of HIV-1 envelope glycoprotein (Env). The VRC01-class bNAb family derived from the VH1-2*02 germline allele arose in multiple HIV-1–infected donors, yet targets the CD4-binding site on Env with common interactions. Modified forms of the 426c Env that activate germline-reverted B cell receptors are candidate immunogens for eliciting VRC01-class bNAbs. We present structures of germline-reverted VRC01-class bNAbs alone and complexed with 426c-based gp120 immunogens. Germline bNAb–426c gp120 complexes showed preservation of VRC01-class signature residues and gp120 contacts, but detectably different binding modes compared to mature bNAb-gp120 complexes. Unlike typical antibody-antigen interactions, VRC01–class germline antibodies exhibited preformed antigen-binding conformations for recognizing immunogens. Affinity maturation introduced substitutions increasing induced-fit recognition and electropositivity, potentially to accommodate negatively-charged complex-type *N*-glycans on gp120. These results provide general principles relevant to the unusual evolution of VRC01–class bNAbs and guidelines for structure-based immunogen design.

## Introduction

The HIV-1 envelope (Env) spike, a trimer of gp120-gp41 heterodimers, is the only target of neutralizing antibodies (Abs). Rapid mutation combined with structural features of the Env trimer that hide conserved features of the HIV-1 spike result in induction of mainly strain-specific neutralizing Abs in most infected individuals. However, broadly neutralizing Abs (bNAbs) evolve in a minority of HIV-1–infected individuals after several years of infection. These Abs are of considerable interest for HIV-1 therapeutic efforts because they can prevent and treat infection in animal models (reviewed in ([*West et al., 2014*]) and exhibited efficacy against HIV-1 in a human clinical trial (*Caskey et al., 2015*). Thus, it is believed that an immunogen that could elicit bNAbs would induce a protective immune response against infection by HIV-1.

Highly potent bNAbs against the conserved CD4-binding site (CD4bs) on gp120 that were derived from the VH1-2*02 variable heavy chain (HC) gene have been isolated from at least nine HIV-1–infected individuals (*Scheid et al., 2011*; *Wu et al., 2010*; *Zhou et al., 2013*; *2015*; *Wu et al., 2011*; *Georgiev et al., 2013*). These bNAbs exhibit unusually high levels of somatic hypermutation, with changes in both complementarity determining regions (CDRs) and framework regions (FWRs) (*Klein et al., 2013*). As exemplified by VRC01, the first such bNAb to be isolated (*Wu et al., 2010*),

**eLife digest** When human immunodeficiency virus-1 (HIV-1) infects humans it can cause a serious disease that damages the immune system. Currently there is no cure for this disease and there are no vaccines available to halt the spread of the virus. Researchers are hoping to be able to develop a single vaccine that can protect individuals against every form (or strain) of HIV-1, but this has proved difficult because many different versions of the virus exist.

An effective vaccine triggers long-lasting immunity to a particular virus or microbe by activating the production of proteins called antibodies that identify and help to destroy the threat. Research has shown that most individuals infected with HIV-1 produce antibodies that can only recognize a few HIV strains. However, there are rare individuals who produce "broadly neutralizing antibodies"; that is, antibodies that can recognize and help to kill 90% or more of HIV-1 strains. Understanding how broadly neutralizing antibodies are produced in infected individuals may aid the development of a vaccine that can protect others from the many circulating strains of HIV.

When an individual encounters a virus, immature antibodies are modified to generate mature antibodies that bind more effectively to specific virus proteins. Here, Scharf et al. investigated how a class of broadly neutralizing antibodies called VRC01-class antibodies, which bind to an HIV protein called gp120, are produced. The experiments used a technique called X-ray crystallography to reveal the three-dimensional structures of immature versions of these antibodies when they are bound to gp120.

Scharf et al. discovered that, unlike most antibodies, the overall final structure of VRC01 antibodies is formed before the antibody matures. Instead of making large changes to the structure of these antibodies, the maturation process makes VRC01-class antibodies become more positively charged, which allows them to bind to gp120 proteins on a wider variety of HIV viruses. These findings suggest that it may be possible to use modified gp120 proteins in vaccines to trigger the production of broadly neutralizing antibodies against HIV.

VRC01-class bNAbs share a common mode of gp120 binding in which the VH1-2*02-derived variable heavy ($V_H$) domain mimics CD4 (*Wu et al., 2010*; *Zhou et al., 2010*; *2013*; *2015*; *Diskin et al., 2011*). Signature features of the HCs of VRC01-class bNAbs that explain their derivation from the VH1-02*02 gene segment include $Trp50_{HC}$, $Asn58_{HC}$, and $Arg71_{HC}$; an additional signature residue, $Trp100B_{HC}$ within CDRH3, is derived from the joining of the D and J gene segments to the VH gene segment during V(D)J recombination (*Diskin et al., 2012*). The variable light ($V_L$) domains of VRC01-class bNAbs, which can be derived from several different VL germline gene segments, require a short (five residues) CDRL3 loop to allow interactions with gp120 V5 and loop D (*Diskin et al., 2012*) and often include a deletion in CDRL1 to avoid clashes with an *N*-linked glycan attached to $Asn276_{gp120}$ (*Zhou et al., 2010*; *2013*; *Scharf et al., 2013*). The LC contact residues of VRC01-like bNAbs and the short CDRL3 are not germline encoded (*Diskin et al., 2012*). Because VRC01-class bNAbs are highly potent, have demonstrated efficacy in animal and human trials, and evolved in multiple HIV-1–infected donors who were infected with different strains of HIV-1, they are considered a promising target for elicitation by vaccination with designed immunogens.

The first step in eliciting an Ab is binding to its precursor germline-encoded B cell receptor by an antigen or immunogen with adequate affinity to initiate B cell activation and affinity maturation (*Batista and Neuberger, 1998*). Although the unmutated VH1-2*02 gene segment includes the signature $V_H$ domain residues for interacting with the CD4bs, germline-reverted versions of VRC01-class bNAbs do not bind to Env trimers or to gp120 (*Scheid et al., 2011*; *Zhou et al., 2010*; *Diskin et al., 2012*; *Scharf et al., 2013*; *Hoot et al., 2013*; *Ota et al., 2012*; *McGuire et al., 2013*). In previous studies, we investigated VRC01-class germline recognition of Env by solving a crystal structure of the clade A/E 93TH057 gp120 core bound to a half germline, half mature chimeric VRC01-class Ab ($NIH45-46_{CHIM}$) in which the inferred germline HC was paired with the mature LC to permit binding to an unmodified gp120 (*Scharf et al., 2013*). Two forms of designed gp120-based immunogens were subsequently engineered to bind and activate germline HC/LC VRC01-class B cell receptors: gp120 outer domain-only constructs (eOD-GT6 and eOD-GT8) (*Jardine et al., 2013*; *2015*) and

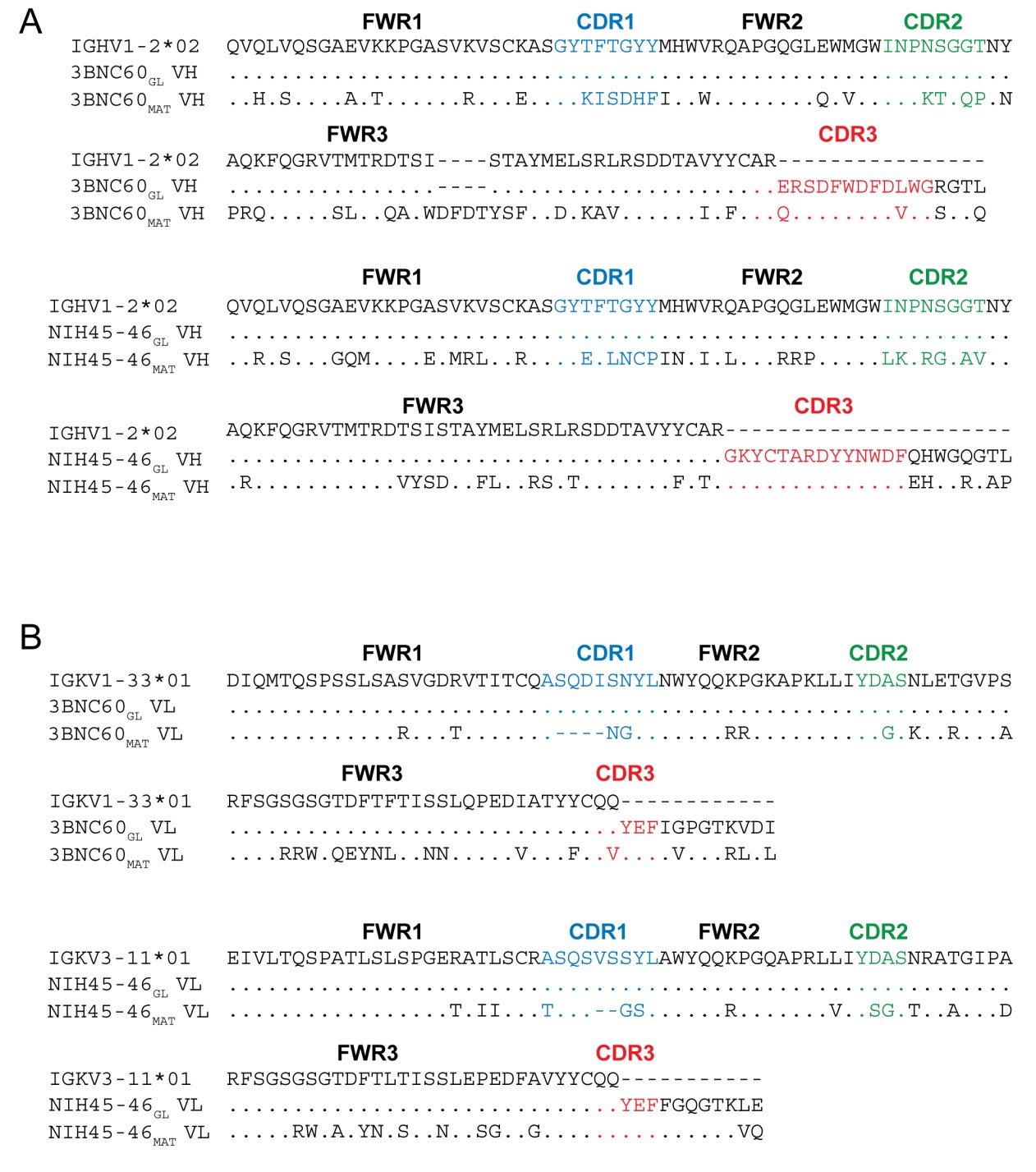

**Figure 1.** Sequence Alignments of Inferred Germline and Mature Forms of 3BNC60 and NIH45-46. Alignments of (**A**) $V_H$ and (**B**) $V_L$ sequences of inferred germline progenitors (3BNC60GL and NIH45-46GL), mature 3BNC60 (3BNC60MAT), mature NIH45-46 (NIH45-46MAT), and the predicted germline V gene segments from which they were derived. Antibody framework regions (FWR) and CDR loops (CDR) are marked and CDR loops are colored blue (CDR1), green (CDR2), and red (CDR3). CDR3 sequences for germline Fabs were taken from mature antibody sequences as done in previous studies (*Dosenovic et al., 2015*; *Hoot et al., 2013*; *McGuire et al., 2013*; *Scharf et al., 2013*).

gp120s modified from the clade C 426c Env (*McGuire et al., 2013*; *2014*; *2016*). Crystal structures of the inferred germline Fab of VRC01 complexed with eOD-GT6 and NIH45-46$_{CHIM}$ complexed with gp120 revealed a similar angle of approach for binding as mature Fabs and conservation of the signature VRC01-class interactions with gp120 residues (*Scharf et al., 2013*; *Jardine et al., 2013*). However, questions concerning germline B-cell receptor recognition of gp120 remained because neither structure represented a complex between a fully germline VRC01-class Ab and a complete gp120 core. Here, we present crystal structures of inferred germline VRC01-like Abs alone and complexed with 426c-based gp120 immunogen cores and compare and contrast their gp120 recognition with structures of mature VRC01-class Ab/gp120 complexes. The analyses shed insight on the evolution pathway by which Abs derived from VH1-2*02 germline mature towards broad recognition of the CD4bs on gp120 and provide structural information that will facilitate design of immunogens to elicit VRC01-class bNAbs.

## Results

### Expression and characterization of germline and mature VRC01-class Abs

The antigen-binding Fabs of two VRC01-class Abs isolated from different patients, NIH45-46 and 3BNC60 (*Scheid et al., 2011*), were generated in their inferred germline forms (NIH45-46$_{GL}$ and 3BNC60$_{GL}$) using sequences described in previous studies (*Scharf et al., 2013*; *Hoot et al., 2013*; *McGuire et al., 2013*; *Dosenovic et al., 2015*). Compared with mature NIH45-46 (NIH45-46$_{MAT}$), NIH45-46$_{GL}$ contained 40 HC and 23 LC substitutions in addition to a two-residue insertion in CDRL1. 3BNC60$_{GL}$ contained 38 HC and 25 LC substitutions in addition to a four-residue deletion in HC framework region 3 (FWR3$_{HC}$) and a four-residue insertion in CDRL1 compared with 3BNC60$_{MAT}$ (*Figure 1*). Two versions of germline-binding gp120s were expressed as gp120 cores with N/C termini and V1-V2 and V3 loop truncations (*Zhou et al., 2010*): (*i*) 426c.NLGS.TM1△V1-3 (hereafter referred to as 426c.TM1△V1-3), a modified version of the clade C 426c Env in which the V1, V2 and V3 loops were truncated and three potential *N*-linked glycosylation sites at gp120 residues Asn276$_{gp120}$, Asn460$_{gp120}$ and Asn463$_{gp120}$ were removed by mutation (N276D, N460D, and N463D) (*McGuire et al., 2013*; *2014*), and (*ii*) 426c.TM4△V1-3, a modification of 426c.TM1△V1-3 with additional substitutions (D276N, S278R, G471S) (*McGuire et al., 2016*; *Dosenovic et al., 2015*). 426c.TM1△V1-3 binds to germline VRC01 and NIH45-46 (*McGuire et al., 2013*; *2014*), and 426c.TM4△V1-3 binds to germline versions of 12A21, 3BNC60, VRC-CH31, VRC-PG19 and VRC-PG20 in addition to germline VRC01 and NIH45-46 (*McGuire et al., 2016*).

We used a surface plasmon resonance (SPR)-based assay to compare the binding of NIH45-46$_{GL}$, 3BNC60$_{GL}$, NIH45-46$_{MAT}$, and 3BNC60$_{MAT}$ to gp120 cores including the 426c immunogens and a representative gp120 (93TH057) used for structural studies with mature bNAbs (*Figure 2*). As expected, the germline-reverted Abs did not bind detectably to 93TH057 gp120, but NIH45-46$_{GL}$ bound to both 426c.TM1ΔV1-3 and 426c.TM4ΔV1-3 gp120s with equilibrium dissociation constants ($K_D$s) of ~0.7 and 3 μM, respectively. As previously described (*McGuire et al., 2013*; *2014*), 3BNC60$_{GL}$ did not bind to 426c.TM1ΔV1-3, but showed detectable binding to 426c.TM4ΔV1-3 gp120. By contrast, NIH45-46$_{MAT}$ and 3BNC60$_{MAT}$ bound to all gp120s with nM or higher affinity (*Figure 2*).

Crystal structures of NIH45-46$_{GL}$ bound to 426c.TM1△V1-3 and 3BNC60$_{GL}$ bound to 426c.TM4△V1-3 were solved to 3.4 Å and 3.1 Å resolution, respectively (*Table 1*). We also solved structures of unbound 3BNC60$_{GL}$ and 426c.TM4△V1-3 (at 1.9 Å and 2.0 Å resolution, respectively), and a complex structure of a 426c.TM4△V1-3 bound to 3BNC55$_{MAT}$, a mature VRC01-class Ab (*Table 1*). We compared the new structures to previously-reported complex structures of NIH45-46$_{CHIM}$/93TH057 gp120 and VRC01$_{GL}$/eOD-GT6 (PDB codes 4JDT and 4JPK), unbound structures of NIH45-46$_{GL}$ and VRC01$_{GL}$ (PDB codes 4JDV and 4JPI) (*Scharf et al., 2013*; *Jardine et al., 2013*), and representative mature VRC01-class Fab/gp120 complexes; e.g., NIH45-46$_{MAT}$/93TH057 gp120 and 3BNC117$_{MAT}$/93TH057 gp120 (PDB codes 3U7Y and 4JPV) (*Zhou et al., 2013*; *Diskin et al., 2011*).

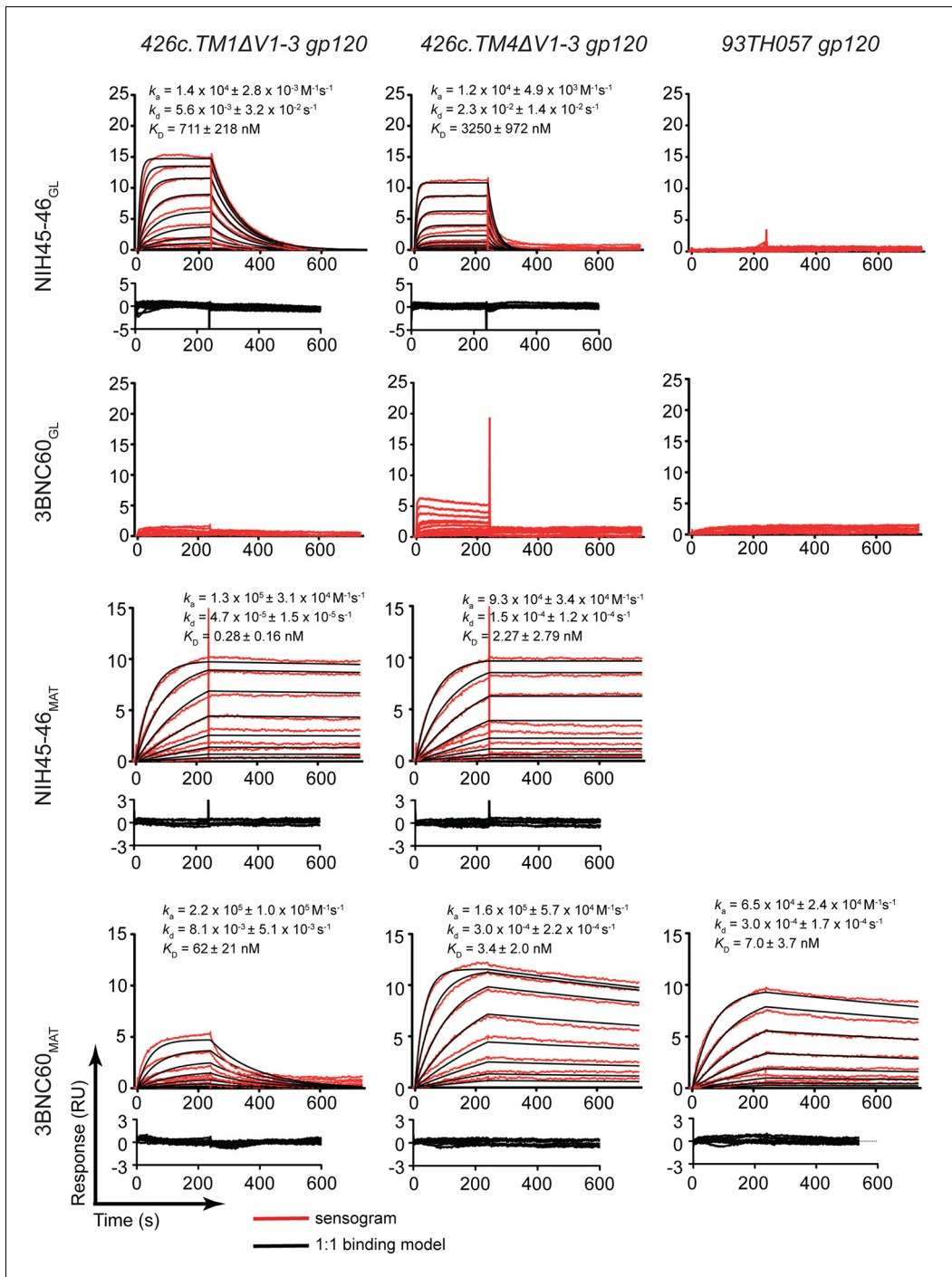

**Figure 2.** SPR binding assays. Representative sensograms (red), fits (black, where applicable), residuals, and $K_D$, $k_a$, and $k_d$ values (mean ± standard deviation from 3 independent experiments) for binding of germline and mature Abs to gp120 cores. NIH45-46$_{GL}$, 3BNC60$_{GL}$, NIH45-46$_{MAT}$ and 3BNC60$_{MAT}$ IgG were captured on a protein A biosensor chip, and 426c.TM1△V1-3, 426c.TM4△V1-3 and 93TH057 gp120 cores were flowed over the chip as a 2-fold dilution series with top concentrations of 16 mM and 400 nM for germline and mature Abs, respectively. DOI: 10.7554/eLife.13783.004

## Germline Abs exhibit preformed antigen-binding conformations

To evaluate the structural plasticity of the inferred germline versions of the VRC01-class bNAbs, we superimposed available structures of inferred germline Fabs in their free and antigen-complexed

**Table 1.** Data collection and refinement statistics, molecular replacement.

| | 3BNC60$_{GL}$ Fab (5F7E) | 426c. TM4ΔV1-3 (5FA2) | 3BNC60$_{GL}$ Fab-426c. TM4ΔV1-3 complex (5FEC) | NIH45-46$_{GL}$ Fab-426c. TM1ΔV1-3 complex (5IGX) | 3BNC55$_{MAT}$ Fab-426c. TM4ΔV1-3 complex (5I9Q) |
|---|---|---|---|---|---|
| Resolution range | 34.13 - 1.9 (1.97 - 1.9) | 35.8 - 2.0 (2.072 - 2.0) | 37.44 - 3.1 (3.211 - 3.1) | 39.37 - 3.4 (3.521 - 3.4) | 37.54 - 2.8 (3.10 - 3.0) |
| Space group | P2$_1$2$_1$2$_1$ | C121 | P2$_1$2$_1$2$_1$ | F222 | P3$_1$21 |
| Unit cell dimensions | | | | | |
| a, b, c (Å) | 74.9, 74.9, 83.1 | 144.9, 85.9, 90.0 | 103.1, 134.1, 195.0 | 147.1, 169.3, 177.7 | 122.9, 122.9, 265.0 |
| α, β, γ (°) | 90.0, 90.0, 90.0 | 90.0, 104.8, 90.0 | 90.0, 90.0, 90.0 | 90.0, 90.0, 90.0 | 90.0, 90.0, 120.0 |
| Total reflections | 243498 (23377) | 416316 (24886) | 335536 (34539) | 207967 (20780) | 710649 (51477) |
| Unique reflections | 37256 (3651) | 81715 (7961) | 49653 (4885) | 15410 (1508) | 57299 (4353) |
| Multiplicity | 6.5 (6.5) | 4.6 (4.5) | 6.1 (6.4) | 13.5 (13.8) | 12.4 (11.8) |
| Completeness (%) | 0.95 (0.98) | 0.97 (0.95) | 1.00 (1.00) | 1.00 (1.00) | 0.71 (95.7) |
| Mean I/sigma(I) | 16.56 (2.36) | 13.35 (0.6) | 9.75 (1.47) | 21.76 (4.03) | 8.65 (1.3) |
| Wilson B-factor | 23.2 | 30.3 | 68.98 | 97.44 | 77.46 |
| R-merge | 0.09029 (0.8891) | 0.06624 (1.36) | 0.1729 (1.305) | 0.1087 (0.6832) | 0.327 (3.774) |
| CC1/2 | 0.999 (0.831) | 0.999 (0.437) | 0.995 (0.483) | 0.999 (0.909) | 0.993 (0.186) |
| CC* | 1 (0.953) | 1 (0.78) | 0.999 (0.807) | 1 (0.976) | 0.998 (0.56) |
| R$_{work}$ | 0.196 | 0.207 | 0.203 | 0.279 | 0.240 |
| R$_{free}$ | 0.210 | 0.232 | 0.267 | 0.286 | 0.279 |
| *Number of atoms* | 3502 | 5860 | 16611 | 5061 | 11572 |
| macromolecules | 3226 | 5124 | 16198 | 4956 | 11258 |
| ligands | | 320 | 413 | 105 | 314 |
| Protein residues | 429 | 672 | 2182 | 697 | 1537 |
| RMS (bonds) | 0.01 | 0.012 | 0.014 | 0.017 | 0.014 |
| RMS (angles) | 1.28 | 1.534 | 1.22 | 1.75 | 1.35 |
| Clashscore | 4.93 | 6.6 | 15.17 | 9.77 | 28.65 |
| *Average B-factor* | 33.0 | 48.66 | 84.92 | 95.14 | 86.29 |
| macromolecules | 31.14 | 47.27 | 83.94 | 93.81 | 86.98 |
| ligands | | 70.79 | 123.27 | 165.32 | 61.48 |
| solvent | 38.67 | 48.77 | | | |

Statistics for the highest-resolution shell are shown in parentheses.

forms. Superimpositions of bound and unbound forms of NIH45-46$_{GL}$ and 3BNC60$_{GL}$ Fabs (this study) and bound and unbound VRC01$_{GL}$ (*Jardine et al., 2013*) revealed no major structural changes (*Figure 3A,C,E*). We previously noted slight differences between bound and unbound NIH45-46$_{MAT}$ in the conformations of CDRs L1 and H3 and positions of the Tyr89$_{CDRL3}$ and Tyr74$_{FWRH3}$ side chains (*Figure 3G*) (*Diskin et al., 2011*), but the counterpart residues in NIH45-46$_{GL}$ were not affected by

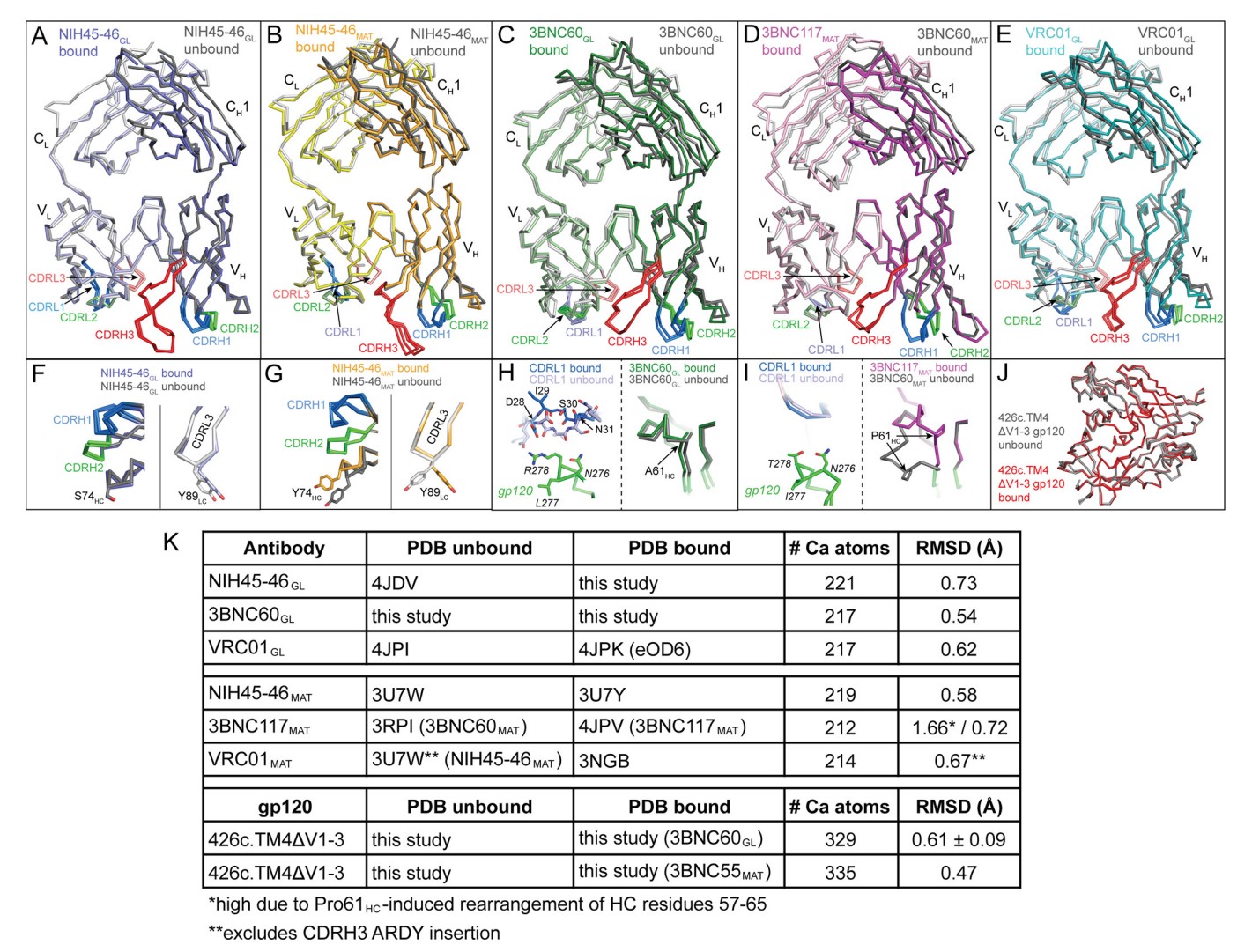

**Figure 3.** Overview of Bound and Unbound Structures of Germline and Mature Forms of NIH45-46 and 3BNC60. Superposition of unbound (grey) and bound (colored) Fab structures of (A) NIH45-46$_{GL}$ (blue), (B) NIH45-46$_{MAT}$ (orange), (C) 3BNC60$_{GL}$ (green), (D) 3BNC60$_{MAT}$ (purple), and (E) VRC01$_{GL}$ (teal). The crystal structures were superimposed on their V$_H$V$_L$ domains and are shown as wire representations with CDR loops colored blue (CDR1), green (CDR2), and red (CDR3). Panels (F–I) show detailed areas of interest for the corresponding structure comparisons shown in panels (A–D). Protein backbones are shown as wire diagrams, side chains are shown as stick representations (red, oxygen; blue, nitrogen). (J) Superposition of unbound (grey) and bound (red) structures of 426c.TM4△V1-3 shown as wire diagrams. (K) Table summarizing Cα rmsds for the indicated Ab and gp120 pairs. Since no unbound crystal structure of 3BNC117$_{MAT}$ was available, the structure of the close clonal relative, 3BNC60$_{MAT}$ (93%HC/96% LC sequence identity) was substituted. Similarly, no unbound structure of VRC01$_{MAT}$ was available, so that of NIH45-46$_{MAT}$ (88%HC/96% LC sequence identity) was substituted, omitting its four-residue insertion in CDRH3 from the rmsd calculation.

gp120 binding (*Figure 3F*). In addition, superimpositions of bound and unbound 3BNC60$_{GL}$ Fab showed no major structural changes except in CDRL1, which either flipped the backbone conformation of the Asp28$_{LC}$-Asn31$_{LC}$ segment (in the case of one Fab/gp120 complex in the crystallographic asymmetric unit; *Figure 3H*) or was partially disordered (in the case of the second Fab/gp120 complex in the asymmetric unit), thus avoiding clashes with loop D of gp120 despite a four-residue insertion in CDRL1. In the mature Ab 3BNC60$_{MAT}$ (and presumably its close relative 3BNC117$_{MAT}$), a disrupted β-strand in the region of Pro61$_{HC}$ was reordered upon binding to gp120 (*Klein et al., 2013*), but no notable differences in the bound and free forms of 3BNC60$_{GL}$ were seen in the counterpart region, which contained Ala61$_{HC}$ instead of Pro61$_{HC}$ (*Figure 3H,I*). Root mean square

deviations (rmsds) for superimposing all Cα atoms in the $V_H V_L$ domains of bound and unbound Fabs of NIH45-46$_{GL}$, 3BNC60$_{GL}$, and VRC01$_{GL}$ were low (0.54 Å–0.73 Å) (*Figure 3K*). Thus the association of the $V_H$ and $V_L$ domains and the conformations of the CDR loops and surrounding FWRs found in the unbound forms were largely maintained in the germline-inferred Abs upon binding to their antigens.

Superpositions of bound and unbound forms of the mature counterparts of these bNAbs (using a closely-related unbound Fab structure when necessary) also revealed relatively small rmsd values with the exception of the 1.66 Å rmsd when comparing free 3BNC60$_{MAT}$ and complexed 3BNC117$_{MAT}$ due to the disrupted β-sheet structure near Pro61$_{HC}$ of free 3BNC60$_{MAT}$ (*Figure 3I*) (*Klein et al., 2013*) – when the disrupted β-strand was excluded, the rmsd dropped to 0.72 Å (*Figure 3K*). Thus the mature bNAbs (NIH45-46$_{MAT}$, 3BNC60$_{MAT}$, and VRC01$_{MAT}$) are overall largely preformed to recognize antigen, but exhibited more antigen binding-induced conformational flexibility within individual CDR loops and/or the FWRs than the germline-inferred Abs.

We addressed potential changes in antigen structure resulting from germline and mature Ab binding by comparing structures of free and bound 426c.TM4△1–3 gp120 core. The 426c.TM4△1–3 structure was solved using crystals of the gp120 core alone and again as part of the analysis of the

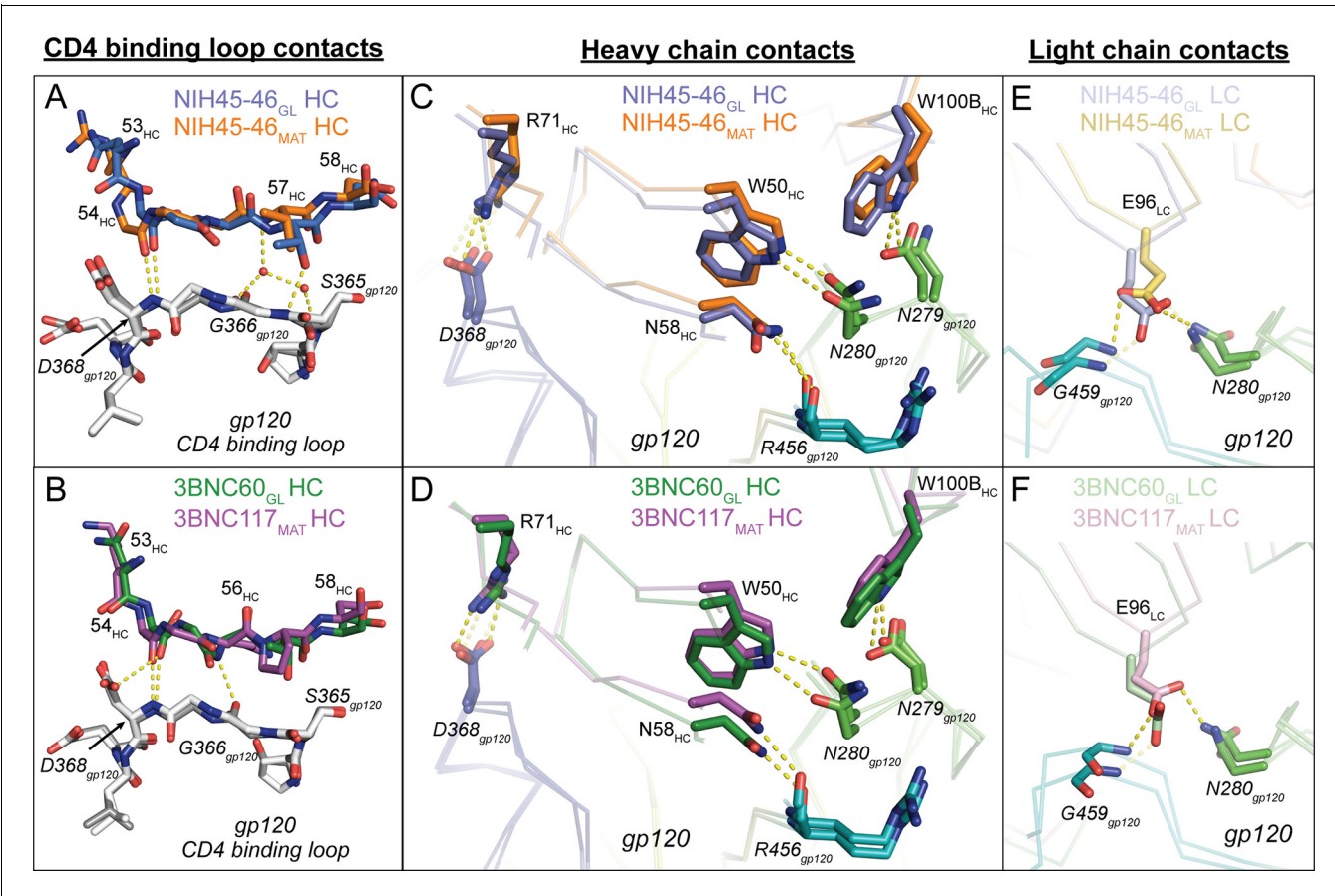

**Figure 4.** Comparison of Signature VRC01-class and CD4-mimicry Contacts in Germline Ab-gp120 Immunogen Complexes. Top panels show (**A**) CD4 binding loop, (**C**) HC, (**E**) LC contacts in superimposed NIH45-46$_{GL}$/426c.TM1△V1-3 and NIH45-46$_{MAT}$/93TH057 (PDB 3U7Y) complexes. Bottom panels show (**B**) CD4-binding loop, (**D**) HC, (**F**) LC contacts in superimposed 3BNC60$_{GL}$/426c.TM4△V1-3 and 3BNC117$_{MAT}$/93TH057 (PDB 4JPV) complexes. Protein backbones are shown as wire diagrams, interacting residues are shown as stick representations (red, oxygen; blue, nitrogen). Yellow dashed lines indicate putative hydrogen bonds (distance < 3.5 Å, A-H–D angle > 90°). Ab coloring: blue, NIH45-46$_{GL}$ HC; light blue, NIH45-46$_{GL}$ LC; orange, NIH45-46$_{MAT}$ HC; yellow, NIH45-46$_{MAT}$ LC; green, 3BNC60$_{GL}$ HC; light green, 3BNC60$_{GL}$ LC; purple, 3BNC60$_{MAT}$ HC; light pink 3BNC60$_{MAT}$ LC. gp120 coloring: blue, CD4-binding loop; green, loop D; teal, loop V5. (E, F) Interacting residues of the C" strand of Fab HCs and the CD4-binding loop of gp120 (grey) are shown as sticks. The Fab residue numbers are indicated since the amino acid sequence differs at some positions between germline and mature Abs.

3BNC60$_{GL}$/426c.TM4△V1-3 crystals in which the crystallographic asymmetric unit included two copies of unbound 426c.TM4△V1-3 gp120 along with two copies of the 3BNC60$_{GL}$/426c.TM4△V1-3 complex. We compared the unbound 426c.TM4△V1-3 structures to those of 426c.TM4△V1-3 complexed with Fabs from 3BNC60$_{GL}$ and from a mature VRC01-class Ab, 3BNC55$_{MAT}$. We found no major structural rearrangements resulting from binding of either a germline or mature Ab (*Figure 3J,K*), suggesting that the 426c gp120 core immunogens are preformed for binding to germline and mature forms of VRC01-class Abs. Since these immunogens lack the V1V2 and V3 loops, potential rearrangement of these regions in the context of the trimer upon binding remains to be studied.

## Germline VRC01-class Fabs recognize gp120 using signature VRC01-class contacts

The new germline Fab/gp120 structures, NIH45-46$_{GL}$/426c.TM1△V1-3 and 3BNC60$_{GL}$/426c. TM4△V1-3, showed interface interactions in which the germline Fab contacts with the CD4-binding loop and loops D and V5 of the gp120 outer domain were similar to those observed for mature VRC01-class Fab/gp120 complexes (*Zhou et al., 2010*; *2015*; *Diskin et al., 2011*). This confirms the assumption of similar recognition modes derived from germline Fab-antigen complex structures that either did not contain a full gp120 core (VRC01$_{GL}$/eOD-GT6) (*Jardine et al., 2013*) or included a mature Ab LC (NIH45-46$_{CHIM}$/93TH057) (*Scharf et al., 2013*). In particular, both NIH45-46$_{GL}$ and 3BNC60$_{GL}$ interacted with the CD4-binding loop on gp120 using conserved interactions that mimic CD4 binding, first described for the complex of VRC01$_{MAT}$ with 93TH057 gp120 (*Zhou et al., 2010*), in which VRC01-class bNAbs mimic CD4 using backbone atoms in the V$_H$ domain C″ strand to engage with the CD4-binding loop on gp120 (*Wu et al., 2010*; *Zhou et al., 2010*; *2015*; *Diskin et al., 2011*). As found in NIH45-46$_{MAT}$ and 3BNC117$_{MAT}$ complexes with 93TH057 gp120s, Gly54$_{HC}$ in NIH45-46$_{GL}$ and 3BNC60$_{GL}$ makes a main chain hydrogen bond with Asp368$_{gp120}$ (*Figure 4A,B*). The 3BNC60$_{GL}$ complex with 426c.TM4△V1-3 gp120 includes an additional main chain hydrogen bond between Gly57$_{HC}$ and Gly365$_{gp120}$ that is not made in the 3BNC117$_{MAT}$/gp120 complex. Similarly, NIH45-46$_{GL}$ Thr58$_{HC}$ makes an additional side chain hydrogen bond with Gly365$_{gp120}$. NIH45-46$_{MAT}$ further engages the CD4 binding loop of gp120 using water-mediated hydrogen bonds between Val57$_{HC}$ and Gly366$_{gp120}$/Asp368$_{gp120}$ (*Diskin et al., 2011*). While the resolution of the germline Fab complex structures did not permit placement of water molecules, residues in the C″ strand of both NIH45-46$_{GL}$ and 3BNC60$_{GL}$ are positioned to engage in water-mediated hydrogen bonds with additional CD4-binding loop residues in gp120. Thus, our analyses showed that NIH45-46$_{GL}$ and 3BNC60$_{GL}$, like their mature counterparts, use the C″ strand in V$_H$ to mimic Leu44$_{CD4}$ and Lys46$_{CD4}$ interactions with the gp120 CD4 binding loop.

We next examined the interactions of the VRC01-class signature residues (*Diskin et al., 2012*). As found for mature VRC01-like Fab/gp120 complexes (e.g., PDB codes 3U7Y and 4JPV) and the half-germline NIH45-46$_{CHIM}$/93TH057 gp120 complex (*Scharf et al., 2013*), the VH1-2*02 germline-encoded heavy chain signature residues (Trp50$_{HC}$, Asn58$_{HC}$, Arg71$_{HC}$) engaged in the predicted interactions with 426c.TM1△V1-3 and 426c.TM4△V1-3 in their complexes with NIH45-46$_{GL}$ and 3BNC60$_{GL}$: i.e., Trp50$_{HC}$ and Asn58$_{HC}$ made hydrogen bonds with Asn280$_{gp120}$ and Arg456$_{gp120}$, respectively, and Arg71$_{HC}$ formed a salt bridge with Asp368$_{gp120}$ to mimic the Arg59$_{CD4}$–Asp368$_{gp120}$ interaction (*Kwong et al., 1998*) (*Figure 4C,D*). The final signature interaction within the heavy chain, CDRH3 residue Trp100B$_{HC}$, also made the predicted hydrogen bonding interaction with Asn279$_{gp120}$. In summary, NIH45-46$_{GL}$ and 3BNC60$_{GL}$ make all predicted HC VRC01-class signature contacts with the CD4-binding loop, the V5 loop, and loop D to bind to gp120.

Potent VRC01-class bNAbs pair with LCs that acquire signature residues, Trp67$_{LC}$ and Glu96$_{LC}$, during somatic hypermutation and/or V-J joining, and are associated with short loops for two of the CDRs: CDRL1 (resulting from a two- to four-residue deletion relative to the germline LC) and a five-residue CDRL3 (*Diskin et al., 2012*). Comparison of the NIH45-46$_{GL}$/426c.TM1△V1-3 and 3BNC60$_{GL}$/426c.TM4△V1-3 structures with their mature counterparts (NIH45-46$_{MAT}$/93TH057 and 3BNC117$_{MAT}$/93TH057; PDB codes 3U7Y and 4JPV) revealed no major structural differences between the LC contacts with gp120 cores (*Figure 4E,F*). The greater number of residues in the CDRL1s in the germline Fab-containing complexes (two and four additional residues in NIH45-46$_{GL}$ and 3BNC60$_{GL}$, respectively) result in wider, rather than longer, loops than their mature counterparts (*Figure 5A*). The wider CDRL1s in the germline Fabs are not extended towards gp120, thereby

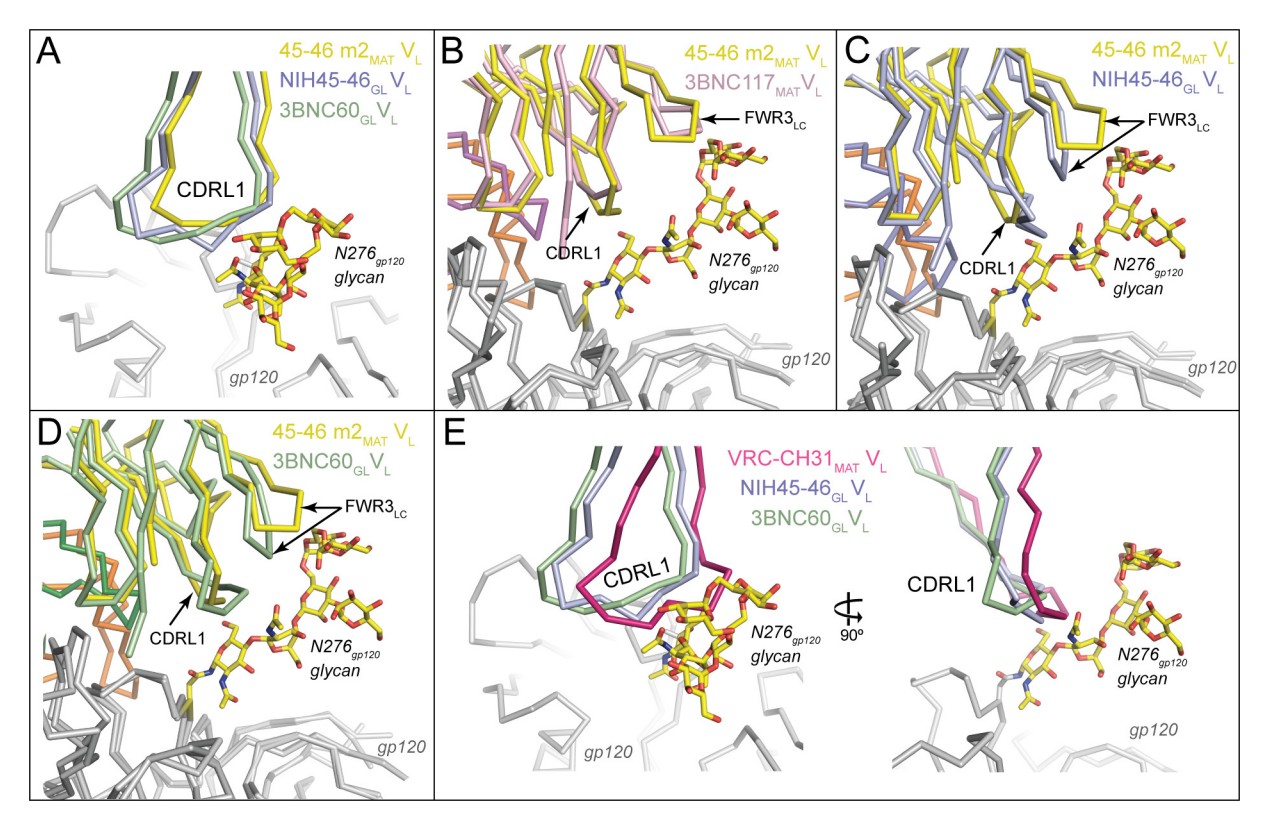

**Figure 5.** Accommodation of Asn276$_{gp120}$ Glycan by Germline Ab Light Chains. Superposition of gp120 (grey) complexes with germline and mature Abs. (**A**) CDRL1 from 45-46m2$_{MAT}$ (yellow), NIH45-46$_{GL}$ (blue) and 3BNC60$_{GL}$ (green) is positioned near the Asn276$_{gp120}$ glycan. Two- and four-residue insertions in NIH45-46$_{GL}$ and 3BNC60$_{GL}$, respectively, result in a widening of the tip of CDRL1 rather than a more extended loop, which would clash with gp120 protein residues and/or the Asn276$_{gp120}$ glycan. gp120 (grey) complexes with (**B**) 3BNC117$_{MAT}$ (pink) and 45-46m2$_{MAT}$ (yellow) (**C**) NIH45-46$_{GL}$ (blue) and 45-46m2$_{MAT}$ (yellow), (**D**) 3BNC60$_{GL}$ (green) and 45–46 m2$_{MAT}$ (yellow). Protein backbones are shown as wire diagrams and the Asn276$_{gp120}$ glycan from the 45-46m2$_{MAT}$/93TH057 gp120 complex (PDB code 4JKP) is shown as sticks (yellow, carbon; red, oxygen; blue, nitrogen). The positions of CDRL1 and FWR3$_{LC}$ are indicated. (**E**) CDRL1 from VRC-CH31$_{MAT}$ (magenta), NIH45-46$_{GL}$ (blue) and 3BNC60$_{GL}$ (green) is positioned near the Asn276$_{gp120}$ glycan. The CDRL1 loop of VRC-CH31$_{MAT}$ is of the same length as that of 3BNC60$_{GL}$, and uses increased backbone conformational flexibility due to somatically mutated glycine residues to avoid clashes with gp120 protein residues and/or the Asn276$_{gp120}$ glycan.

preventing clashes with the 426c.TM1△V1-3 and 426c.TM4△V1-3 gp120s (*Figure 5A*). However, CDRL1 is near the position of the Asn276$_{gp120}$*N*-linked glycan (removed by mutagenesis in gp120-based immunogen candidates including the 426c.TM1△V1-3 and 426c.TM4△V1-3 gp120s and the eOD-GT gp120 outer domains) (*McGuire et al., 2013*; *2014*; *2016*; *Jardine et al., 2013*; *2015*). To address whether this glycan would clash with the larger CDRL1 loops of germline VRC01-class Abs, we superimposed the NIH45-46$_{GL}$/426c.TM1△V1-3 and 3BNC60$_{GL}$/426c.TM4△V1-3 complex structures with the structure of a mature Ab/gp120 complex that includes a partially ordered Asn276$_{gp120}$ glycan (the structure of 45-46m2 complexed with 93TH057 gp120 core; PDB code 4JKP) (*Diskin et al., 2013*). The superimposed germline Fabs showed no clashes between CDRL1 and ordered Asn276$_{gp120}$ glycan residues (*Figure 5B–D*). However, given the flexibility of *N*-linked glycans, some conformations of the Asn276$_{gp120}$-linked glycan could interfere with binding germline CD4bs Abs. VRC01-class Abs VRC-CH31 and 12A21 do not have deletions in CDRL1, but accommodate the Asn276$_{gp120}$-linked glycan due to a pair of glycines that increase the conformational flexibility of CDRL1 (*Zhou et al., 2013*). The CDRL1 of 3BNC60$_{GL}$ is of the same length as the VRC-CH31 and 12A21 CDRL1s and lacks glycines, yet has the conformational flexibility to avoid clashes with loop D$_{gp120}$ and the Asn276$_{gp120}$ glycan (*Figure 5E*). Therefore, CDRL1 deletions or enhanced loop flexibility due to somatically substituted glycine residues is not required for binding to gp120

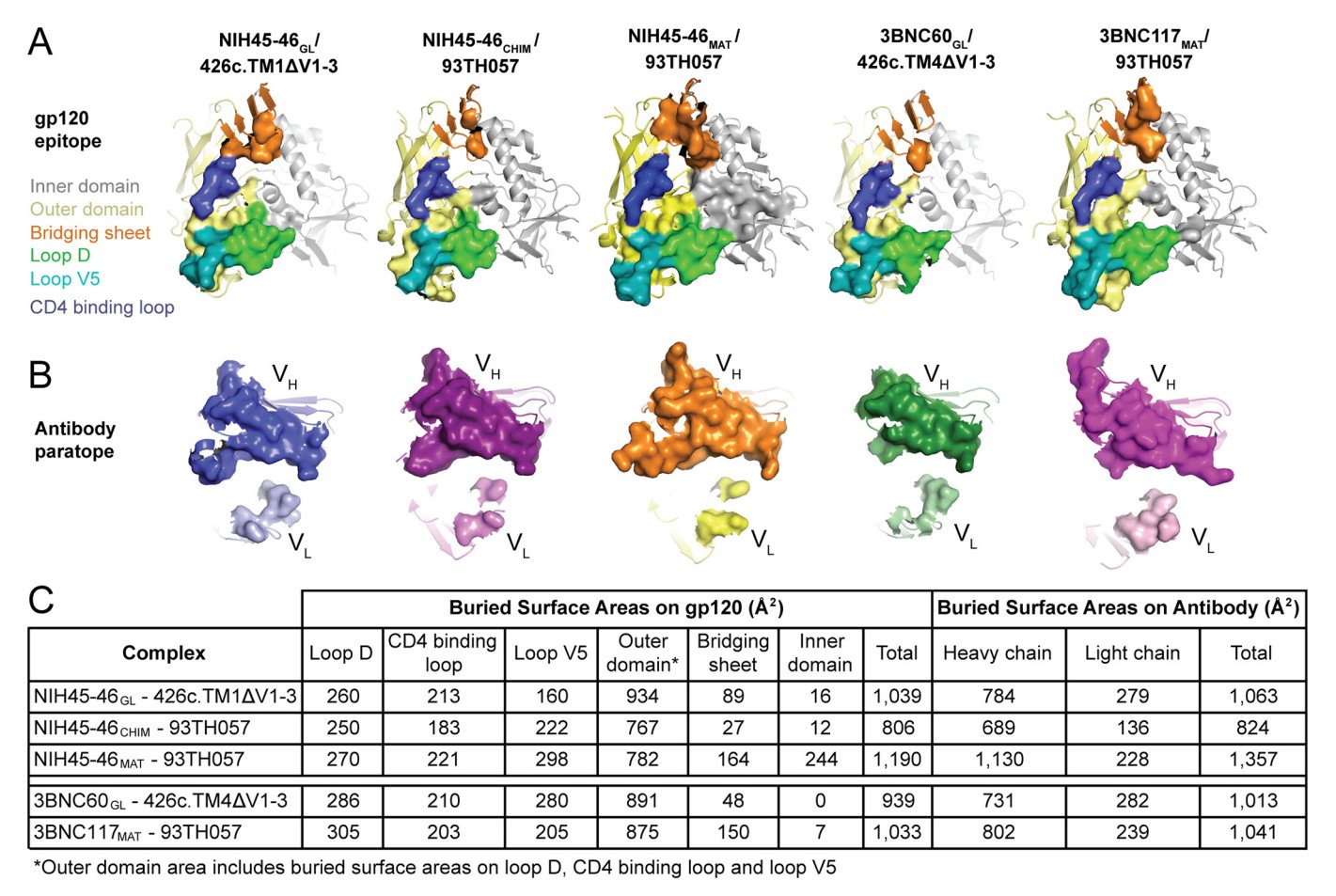

**Figure 6.** Comparison of the Binding Interfaces in gp120 Complexes with Germline and Mature Abs. (**A**) gp120 residues contacted by Fabs and (**B**) Fab residues contacted by gp120s are shown as surfaces over ribbon diagrams. gp120 domains are colored as in *Figure 3*. Ab coloring: blue, NIH45-46$_{GL}$ HC; light blue, NIH45-46$_{GL}$ LC; purple, NIH45-46$_{CHIM}$ HC; light purple, NIH45-46$_{CHIM}$ LC; orange, NIH45-46$_{MAT}$ HC; yellow, NIH45-46$_{MAT}$ LC; green, 3BNC60$_{GL}$ HC; light green, 3BNC60$_{GL}$ LC; pink, 3BNC60$_{MAT}$ HC; light pink, 3BNC60$_{MAT}$ LC. (**C**) Quantitation of buried surface areas (Å$^2$) depicted in (**A**) and (**B**). The columns labeled total are the sums of areas for outer domain, bridging sheet and inner domain for gp120, and of heavy chain and light chain for Abs. Surface areas buried due to complex formation were calculated using a 1.4 Å probe.

although these adaptations likely lead to improved affinity in mature bNAbs adapted to bind to HIV-1 Envs that are not optimized to accommodate germline Ab binding.

## Comparison of interface buried surface areas for germline and mature Fab complexes with gp120 immunogens

We previously noted that the buried surface area on both the Ab and the gp120 was larger in mature Fab/gp120 complexes than in the NIH45-46$_{CHIM}$/93TH057 gp120 complex (*Scharf et al., 2013*). Here, we extended this analysis to include comparisons with the HC/LC germline Fab complexes with 426c.TM1△V1-3 and 426c.TM4△V1-3 (*Figure 6A,B*, *Supplementary file 1*). The surface area buried on gp120 by both germline Fabs was only slightly smaller than that buried by the corresponding mature Abs (1,039 Å$^2$ vs. 1,190 Å$^2$ for NIH45-46$_{GL}$ vs. NIH45-46$_{MAT}$ and 939 Å$^2$ vs. 1,033 Å$^2$ for 3BNC60$_{GL}$ vs. 3BNC117$_{MAT}$), with gains in buried surface area in loop D, inner domain and bridging sheet residues of gp120 (*Figure 6C*). The surface area buried on the Fab HCs by gp120 was smaller than that buried by the corresponding mature Abs with no change in the area buried on the LCs (784 Å$^2$ vs. 1,130 Å$^2$ for NIH45-46$_{GL}$ HC vs. NIH45-46$_{MAT}$ HC and 731 Å$^2$ vs. 802 Å$^2$ for

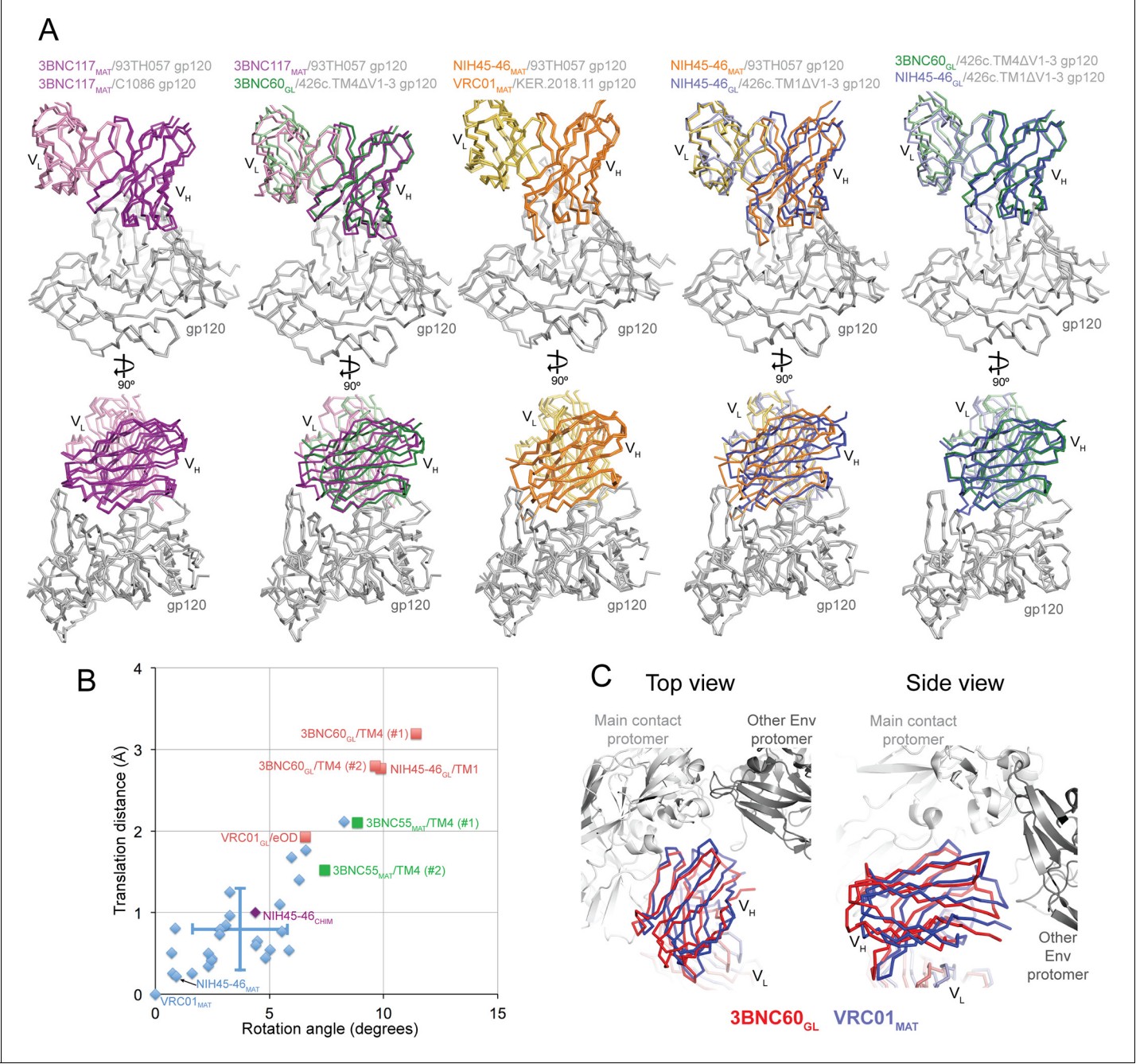

**Figure 7.** Comparisons of Binding Mode in Germline Ab-gp120 Immunogen Complexes. (**A**) Superpositions of Fab-gp120 complexes depicted as wire diagrams. The following Fab/gp120 complexes were compared by alignment of their gp120s: 3BNC117$_{MAT}$/93TH057 (PDB code 4JPV), 3BNC117$_{MAT}$/C1086 (PDB code 4LSV), 3BNC117$_{MAT}$/93TH057 (PDB code 4JPV), 3BNC60$_{GL}$/426c.TM4△V1-3, NIH45-46$_{MAT}$/93TH057 (PDB code 3U7Y), VRC01$_{MAT}$/KER.2018.11 (PDB code 4LSS), NIH45-46$_{GL}$/426c.TM1△V1-3. (**B**) Rotation angle and translation distance of V$_H$ domains of mature, chimeric and germline Fabs in complex with gp120s relative to VRC01$_{MAT}$ in complex with 93TH057 gp120 (PDB code 3NGB). Data points for complexes of mature Fabs bound to non-immunogen gp120s are shown as blue diamonds, complexes of germline Fabs bound to immunogen candidates are shown as red squares (TM1 = 426c.TM1△V1-3, TM4 = 426c.TM4△V1-3, eOD = eOD-GT6), the complex between the half mature, half germline NIH45-46$_{CHIM}$ and to a non-immunogen gp120 is shown as a purple diamond, and 3BNC55$_{MAT}$ bound to 426c.TM4△V1-3 is shown as a green square. When two complexes were found in the crystallographic asymmetric unit, rotation and translation parameters are shown for both complexes (denoted as #1 and #2). Standard deviations for the translation distance and rotation angle for mature VRC01-class bNAb–gp120 complexes shown as vertical and horizontal lines, respectively. (**C**) Alignment of the 3BNC60$_{GL}$/426c.TM4△V1-3 (V$_H$V$_L$ shown in red) and VRC01$_{MAT}$ Fab/gp120 (PDB code 3NGB) (V$_H$V$_L$ shown in blue) structures onto the gp120 region of a native-like Env trimer structure (BG505 SOSIP.664; PDB code 5CEZ) (gray). Modeled structures are shown looking down the trimer three-fold axis (left panel) and from the side (right panel).

*Figure 7 continued on next page*

*Figure 7 continued*

The following source data is available for figure 7:

**Source data 1.** Rotation angle and translation distance data of $V_H$ domains of mature, chimeric and germline Fabs in complex with gp120s relative to VRC01$_{MAT}$ in complex with 93TH057 gp120.

3BNC60$_{GL}$ vs. 3BNC117$_{MAT}$). The difference was most pronounced for the NIH45-46$_{GL}$/NIH45-46$_{MAT}$ comparison, in which the mature Fab gained contacts in CDRH3.

## Differences in approach angle for Abs complexed with 426c-based gp120 immunogens

Mature VRC01-class Abs show similar angles of approach for binding to gp120 ([Zhou et al., undefined] and references therein). However, comparison of the orientations of the germline Fab interactions with gp120 in the NIH45-46$_{GL}$/426c.TM1△V1-3 and 3BNC60$_{GL}$/426c.TM4△V1-3 structures revealed differences compared with their mature Fab counterparts (*Figure 7A*). To systematically analyze these differences, we calculated the rotation and translation for the $V_H$ domains of mature, chimeric, and germline Fabs bound to gp120s when compared with a reference Fab/gp120 structure, the VRC01$_{MAT}$/93TH057 complex (*Zhou et al., 2010*) (PDB code 3NGB) (*Figure 7B*). We found that the mature Fab/gp120 complex structures clustered in the recognition mode for VRC01-class Ab recognition of gp120 (Zhou et al., undefined). The germline Fab complexes with the 426c gp120 immunogens exhibited larger rotations and translations relative to the VRC01 reference structure, with the VRC01$_{GL}$/eOD-GT6 complex showing an intermediate orientation. To verify that the orientations observed for the germline Fabs bound to the 426c gp120s are relevant for binding to Env trimer, we aligned the 3BNC60$_{GL}$/426c.TM4△V1-3 structure onto the gp120 region of a native-like Env trimer structure (BG505 SOSIP.664; PDB code 5CEZ) (*Garces et al., 2015*), comparing this orientation with the alignment of a VRC01$_{MAT}$/gp120 complex (PDB code 3NGB) (*Zhou et al., 2010*) onto the same Env trimer structure (*Figure 7C*). This alignment suggests that the different orientation of the 3BNC60$_{GL}$ $V_H$ domain with respect to gp120 moves it away from the adjacent gp120 subunit, suggesting this orientation would be sterically compatible when binding to Env trimer.

The orientation differences for the NIH45-46$_{GL}$/426c.TM1△V1-3 and 3BNC60$_{GL}$/426c.TM4△V1-3 structures could result from structural characteristics of germline Fabs, immunogen characteristics allowing for germline Fab recognition, or a combination of these factors. Arguing in favor of the idea that the 426c immunogens could select for an altered binding mode, we note that a mature Ab (3BNC55$_{MAT}$) complex with 426c.TM4△V1-3 exhibited rotation and translation values closer to those of the germline Fab/426c immunogen complexes than those of mature Ab-gp120 complexes. In addition, the VRC01$_{GL}$ complex with another germline-binding immunogen, eOD-GT6, exhibited a binding mode that differed from the VRC01 reference complex more than most of the mature Fab/gp120 complexes. We also note that the half germline NIH45-46$_{CHIM}$ Fab clustered with the mature Fab/gp120 complexes when bound to the 93TH057 gp120 (*Figure 7B*). This is consistent with the idea that designed immunogens capable of binding germline VRC01-class bNAbs select for a slightly different antibody-binding mode than that adopted by mature VRC01-class Fabs bound to gp120s not capable of supporting germline antibody binding.

## Interactions with complex-type *N*-glycans may be facilitated by electrostatic changes in Fab combining sites during maturation

To address potential global changes in VRC01-class bNAbs during maturation from germline to mature forms, we calculated electrostatic potentials of the binding surfaces of germline and mature Fabs. Comparisons of electrostatic surface potentials revealed a striking shift to more positively-charged antigen combining sites due to maturation (*Figure 8*). The more negatively-charged (3BNC60$_{GL}$) or neutral (NIH45-46$_{GL}$, VRC01$_{GL}$) properties of the antigen-binding sites of the germline Fabs may interfere with interactions with complex-type *N*-glycans on gp120 containing negatively-charged sialic acids; in particular, the Asn276$_{gp120}$ carbohydrate, typically a complex-type *N*-glycan (*Binley et al., 2010*; *Go et al., 2011*), would likely make unfavorable interactions with the neutral or negatively charged surfaces of 3BNC60$_{GL}$, NIH45-46$_{GL}$, or VRC01$_{GL}$ Fabs. Such interactions were

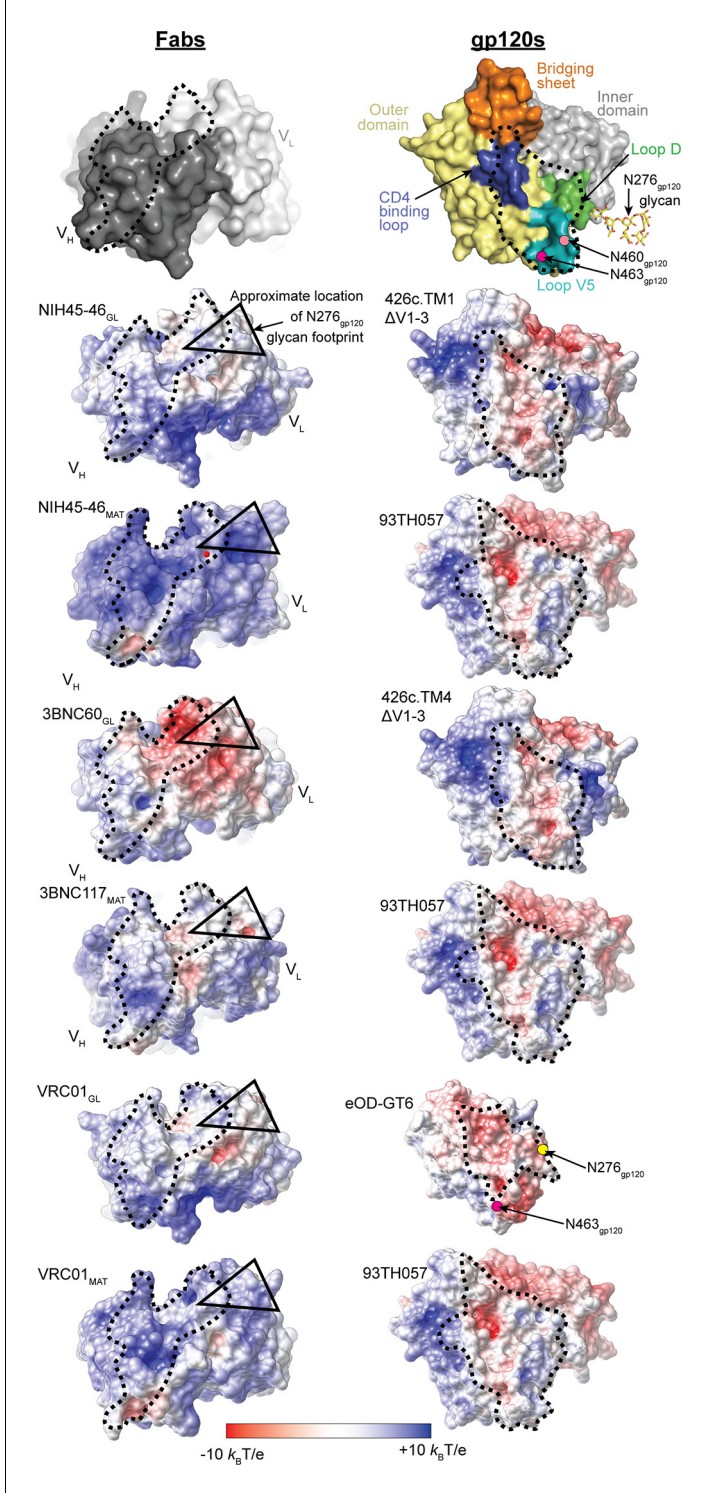

**Figure 8.** Comparison of Electrostatic Surface Characteristics of Fab-gp120 Complexes. The binding surfaces of Fabs (left panels) and gp120s (right panels) are shown. Each binding partner is shown in an orientation looking into the binding interface; the corresponding complex would be obtained by rotating each binding partner by ~90° about the vertical axis. The top panel shows the locations of landmarks on surface representations of Fabs (dark grey, $V_H$; light grey, $V_L$) and gp120s (yellow, outer domain; grey, inner domain; orange, bridging sheet; blue, CD4-binding loop; green, loop D; teal, loop V5; ordered residues of the $Asn276_{gp120}$ glycan shown as yellow sticks (normally a complex-type *N*-glycan, but a high mannose *N*-glycan in crystal structures); approximate locations of $Asn460_{gp120}$ and $Asn463_{gp120}$ shown as light pink and magenta dots, respectively). The lower panels show

*Figure 8 continued on next page*

*Figure 8 continued*

electrostatic potentials on surface representations of Fabs (left panels) and gp120s (right panels) colored blue (positive electrostatic potential) to red (negative electrostatic potential). The binding interfaces are outlined with a dotted black line. The approximate footprints of the complex-type $Asn276_{gp120}$ glycan on Fab surfaces are indicated with a black triangle (the $Asn463_{gp120}$ glycan, also complex-type, is not resolved in any Env structures, thus its footprint on Fab surfaces cannot be shown).

The following figure supplement is available for figure 8:

**Figure supplement 1.** The combining sites of germline (left panels) and mature Fabs (right panels) are shown as surface representations and electrostatic potentials are indicated using blue for positive electrostatic potential and red for negative electrostatic potential.

prevented in the 426c gp120- and eOD-based immunogens by mutation of the $Asn276_{gp120}$ $N$-linked glycosylation sequon (*McGuire et al., 2013*; *2014*; *Jardine et al., 2013*; *2015*). Similarly, potentially unfavorable electrostatic interactions between the germline Fabs and the $Asn463_{gp120}$ $N$-linked glycan, also usually a complex-type $N$-glycan (*Binley et al., 2010*; *Go et al., 2011*), were prevented by mutation of the $Asn463_{gp120}$ glycosylation sequon in the 426c gp120- and eOD-based immunogens (*McGuire et al., 2013*; *2014*; *2016*; *Jardine et al., 2013*; *2015*).

## Discussion

The process by which B cells produce high-affinity Abs starts with antigen binding to an unmutated, germline B cell receptor produced by random joining of V, D, and J or V and J gene segments. Affinity maturation is the process by which Abs with higher antigen-binding affinity are created through somatic hypermutation of the germline B cell receptor (*Victora and Nussenzweig, 2012*). Many anti-HIV-1 bNAbs, including VRC01-class CD4bs bNAbs, are heavily somatically mutated, likely through successive rounds of hypermutation and selection of B cells in response to rapid mutation of HIV-1 Env (*West et al., 2014*; *Klein et al., 2013*). Mature VRC01-class bNAbs achieve nucleotide

**Table 2.** Nominal net charges of VRC01-class bNAbs and of the human Ab repertoire. Nominal net charge is the total charge of the sequence calculated with Asp/Glu as -1 and Arg/Lys as +1. The human Ab repertoire average is calculated from 601,889 HC and 206,953 LC sequences reported in (*Rubelt et al., 2012*). Light chain Vgene assignments are from (*Zhou et al., 2015*). Standard deviations are indicated, except for human Ab repertoire $V_H + V_L$, which cannot be calculated with unpaired chains. *Two-tailed T test comparing the net charges of VRC01-class VHs with human repertoire $V_H$s gives a p-value of 0.01.

| | Nominal Net Charge | | | |
|---|---|---|---|---|
| Ab | $V_H + V_L$ | $V_H$ | $V_L$ | LC V-gene |
| VRC01 | 5 | 4 | 1 | *K3-20* |
| NIH45-46 | 9 | 8 | 1 | *K3-20* |
| 3BNC117 | 5 | 2 | 3 | *K1-33* |
| 12A12 | 5 | 1 | 4 | *K1-33* |
| VRC-PG04 | 7 | 3 | 4 | *K3-20* |
| VRC-CH31 | 2 | 2 | 0 | *K1-33* |
| VRC-PG20 | 4 | 5 | -1 | *L2-14* |
| VRC23 | 5 | 4 | 1 | *K3-15* |
| VRC18 | 3 | 2 | 1 | *K3-20* |
| VRC27 | 4 | 6 | -2 | *K1-33* |
| Average of VRC01-class bNAbs | 4.9 ± 1.9 | 3.7* ± 2.1 | 1.2 ± 1.9 | |
| Human Ab repertoire average | 3.1 | 1.5 ± 2.4 | 1.6 ± 2.1 | |

somatic mutation frequencies of 32% and 20% in HC and LC variable region genes, respectively, whereas the mutation frequencies of more typical affinity-matured human Abs rarely exceeds 10% (*Kwong and Mascola, 2012*). Although heavily somatically mutated, VRC01-class bNAbs are promising targets for vaccine design because they have evolved in multiple donors from a common human germline $V_H$ gene segment, VH1-2*02, to recognize HIV-1 Env using conserved interactions (*Scheid et al., 2011*; *Wu et al., 2010*; Zhou et al., undefined; *Zhou et al., 2013*; *Diskin et al., 2011*; *Zhou et al., 2010*).

The first step in targeted attempts to raise VRC01-class bNAbs by immunization requires identification of an immunogen that binds to the germline configuration of the B-cell receptor, but germline-reverted versions of VRC01-like bNAbs generally do not bind HIV-1 Env (*Scheid et al., 2011*; *Zhou et al., 2010*; *Diskin et al., 2012*; *Scharf et al., 2013*; *Hoot et al., 2013*; *Ota et al., 2012*; *McGuire et al., 2013*; *Jardine et al., 2013*). Although antigens designed to bind to predicted unmutated germline precursors of VRC01-class bNAbs have been constructed and evaluated (*McGuire et al., 2013*; *2014*; *2016*; *Jardine et al., 2013*; *2015*; *Dosenovic et al., 2015*), structural comparisons of germline and mature versions of VRC01-class bNAbs bound to gp120-based immunogens were not available. Here, we report the results of structural studies of immunogens that bind to the inferred germline precursors of VRC01-class bNAbs, discovering both general principles and details of interactions that can guide structure-based immunogen design.

A surprising finding revealed by our structural comparisons of germline and mature VRC01-class Fabs was an increase in electropositive surface potential in the antigen-combining site for the mature Fab compared with the germline-inferred Fab, which we observed in the three VRC01-class Abs (NIH45-46, 3BNC60, and VRC01) for which germline and mature Fab structures were available for calculating electrostatic potentials. This marked shift to a more electropositive antigen combining site may also occur in other VRC01-class bNAbs, particularly those whose LCs are derived from the LC germline *KV1-33* gene segment. Mature VRC01-class bNAbs appear on average to have a higher net nominal charge of their $V_H V_L$ domains (+4.9) than the average human Ab (+3.1; as calculated from sequenced human B-cell repertoires) (*Table 2*) (*Rubelt et al., 2012*), and the VH1-2*02 gene segment yields a sequence with a nominal net charge of +5, higher than the average human $V_H$ gene segment (+2 to +3). However, the *KV1-33* LC germline gene segment, used by some VRC01-class Abs, yields a protein sequence with a nominal net charge of -4, in contrast to the *KV3-20* gene segment (utilized by other VRC01-class Abs) with a net charge of zero. Perhaps to compensate for these negative charges, after maturation of the *KV1-33*–derived VRC01-class bNAbs 3BNC117, VRC-CH31, and 12A12, the portions of the $V_L$ domain encoded by the *KV1-33* gene segment have net nominal charge changes of +5, +3, and +6, respectively.

The trend towards increased electropositivity with Ab maturation does not apply to all HIV-1 Env-specific Abs: maturation of CH58, a non-neutralizing Ab against the gp120 V2 loop that was raised by vaccination in the RV144 trial (*Liao et al., 2013*), did not involve a shift to increased electropositive character (*Figure 8—figure supplement 1*), consistent with its epitope being centered on a positive residue, Lys169$_{gp120}$, within the gp120 V2 loop (*Nicely et al., 2015*). However, the developmental pathway of potent V1V2-directed bNAbs involves increased positivity of the CDRH3 loop, which starts as highly electronegative in part due to sulfated tyrosines, but accumulates positive charges during maturation that partially compensate for the negatively-charged sulfates (*Doria-Rose et al., 2014*). A trend towards electropositive combining sites in mature HIV-1 bNAbs could rationalize their tendencies towards polyreactivity (binding more than one antigen), which has been observed at higher frequencies for broadly neutralizing, as compared with non-neutralizing, Abs against HIV-1 (*Liu et al., 2015*). For example, increased non-specific binding to cardiolipin and nucleic acids, negatively-charged antigens commonly used in polyreactivity assays, correlates with polyreactive properties of Abs (*Mouquet et al., 2010*).

We suggest that evolution of increased electropositivity during maturation of VRC01-class bNAbs (and perhaps other HIV-1 bNAbs; *Figure 8—figure supplement 1*) facilitates interactions with heavily glycosylated HIV-1 Env spikes that include complex *N*-linked glycans containing negatively charged sialic acids. Indeed, somatic hypermutation to increase electropositivity of an Ab combining site may be a strategy utilized by Abs against other viruses containing heavily glycosylated Env proteins, such as influenza and Ebola. In the case of VRC01-class bNAbs, this strategy may be part of a multi-pronged approach of the humoral immune response against the CD4 binding site of HIV-1 for accommodating and/or for avoiding the *N*-linked Env glycan attached to Asn276$_{gp120}$, a site that

can include complex-type *N*-glycans (*Behrens et al., 2016*), with other strategies including deletions to shorten CDRL1 and increasing CDRL1 flexibility by somatic mutations to glycine (*Zhou et al., 2013*). In addition to these steric considerations for potentially unfavorable interactions between CDRL1 and the $Asn276_{gp120}$-linked glycan, the neutral or negatively charged character of the germline Fab combining sites provides a structural justification for the necessity to eliminate the $Asn276_{gp120}$- and $Asn463_{gp120}$-linked *N*-glycans from eOD- and gp120-based immunogen candidates to achieve binding to inferred germline B-cell receptors (*McGuire et al., 2013*; *Jardine et al., 2013*; *2015*; *McGuire et al., 2014*; *2016*). The structures presented here, taken together with immunogen requirements for achieving germline Ab binding and previous structural analyses of mature VRC01-class Abs (Zhou et al., undefined; *Zhou et al., 2010*; *2013*; *McGuire et al., 2013*; *2014*; *2016*; *Jardine et al., 2013*; *2015*), suggest that the initiating antigen(s) in a natural pathway to induce VRC01-class Abs in HIV-1–infected individuals involve viral variants that lack the $Asn276_{gp120}$ (5% of HIV-1 strains in the Los Alamos Database) and the $Asn463_{gp120}$*N*-glycans (~80% of HIV-1 strains in the Los Alamos Database) (http://www.hiv.lanl.gov/content/index). Alternatively, a natural VRC01-class eliciting antigen could be the result of glycan heterogeneity within a single HIV-1 strain that produced variants containing high mannose, rather than complex-type, *N*-glycans at $Asn276_{gp120}$ and/or $Asn463_{gp120}$, consistent with observations of glycan heterogeneity at individual *N*-linked glycosylation sites within single HIV-1 strains (*Go et al., 2011*; *Behrens et al., 2016*).

Although antibody-antigen recognition modes represent a continuum of binding mechanisms, germline precursors to affinity-matured Abs have been suggested to display structural flexibility to allow expanded antigen recognition through induced fit mechanisms, whereas affinity maturation through somatic hypermutation was assumed to convert recognition to a lock-and-key model (*Foote and Milstein, 1994*; *Wedemayer et al., 1997*; *Thorpe and Brooks, 2007*). Induced fit modes of Ab-antigen recognition involve changes in the conformations of backbone and sidechain atoms of both the Ab and the antigen (*Blackler et al., 2011*; *Li et al., 2007*; *Rosen et al., 2005*; *Sinha and Smith-Gill, 2005*; *Verdaguer et al., 1996*), with large changes in the Ab usually occurring in the CDR loops. By contrast, lock-and-key recognition involves a minimum of conformational changes between the bound and unbound states of the antigen and Ab. Here, we showed that for the VRC01 class of CD4bs bNAbs, the maturation pathway for antigen binding from germline Ab to affinity-matured Ab does not involve changes from induced fit to lock-and-key binding mechanisms. Instead, the 426c gp120 immunogens bind $NIH45-46_{GL}$ and $3BNC60_{GL}$ without requiring notable conformational changes in either the Ab or the antigen; thus both are largely preformed for binding. The largely lock-and-key binding mechanism for germline VRC01-class recognition of antigen may be rationalized by the fact that germline VRC01-class CD4bs Abs face major steric constraints when binding to HIV-1 Env trimer due to the recessed location of the epitope and its shielding by glycans from the target and adjacent gp160 monomers (*West et al., 2014*; *Zhou et al., 2015*). VRC01-like CD4bs bNAbs bind Env at an acute angle using mostly CDRs and FWRs of $V_H$ in part because these steric constraints likely do not allow a Fab to bind perpendicular to the CD4bs on a closed Env trimer without clashes with the adjacent trimer subunit. These constraints may also not permit bNAbs or their germline precursors to undergo the major conformational changes characteristic of induced-fit binding and instead select for rare Ab sequences that are preformed to recognize their epitope in this highly sterically-constrained setting. The lock-and-key binding mechanism of VRC01-class recognition also may reflect that the binding interaction is dominated (relative to other Abs) by $V_H$ domain FWRs, and that these would be unlikely to undergo large induced-fit conformational changes.

Structural analyses of germline and mature versions of the non-neutralizing HIV-1 Ab CH58 showed a more typical path for Ab-antigen recognition; the germline precursor used a combination of induced fit and lock-and-key binding modes to recognize a V2 peptide antigen: its CDRL2 loop was preformed for binding but CDRL3 changed conformation upon antigen binding (*Nicely et al., 2015*). The conformation of CDRL3 became fixed in mature CH58 through limited somatic mutation compared with VRC01-class Abs (11 total substitutions in CH58 $V_H V_L$) (*Nicely et al., 2015*). Thus, the conversion from induced fit to lock-and-key binding may be applicable for HIV-1 Abs such as CH58 that include the relatively small numbers of somatic mutations typical for most Abs, but not for VRC01-class Abs, whose maturation process involves a much higher number (>60) of mutations.

The finding of what can be described as lock-and-key recognition for germline Fabs binding to the 426c gp120 immunogens contrasts with recognition of unmodified gp120s used for structural studies of complexes with mature VRC01-class bNAbs that do not bind or bind only poorly to

germline Abs. For example, when bound to the unmodified 93TH057 gp120, the CDRH3 of the germline HC in the NIH45-46$_{CHIM}$-93TH057 complex structure was partially disordered, presumably to avoid clashes with a gp120 not optimized for binding to germline Abs (*Scharf et al., 2013*). An example of structural rearrangements in a mature VRC01-class bNAb bound to an unmodified gp120 is illustrated by a study of the Ala61$_{HC}$Pro substitution in the C″ β-strand of the β-sheet framework in the V$_H$ domains of the 3BNC60/3BNC117 HIV-1 bNAbs. This substitution is required for maximal neutralization potency of the mature bNAbs, yet it disrupts the C″ β-strand of the V$_H$ domain in the unbound Fab and decreases its thermal stability (*Klein et al., 2013*), with the β-sheet framework being restored through a large conformational change when the mature Fab binds gp120 (*Zhou et al., 2013*; *Klein et al., 2013*). Taken together, the findings of what appears to be lock-and-key style germline Fab recognition with increased flexibility upon maturation for the highly somatically-mutated VRC01-class bNAbs contradict the assumption of induced fit for germline Ab recognition and lock-and-key fit for mature Ab binding (*Foote and Milstein, 1994*; *Wedemayer et al., 1997*; *Thorpe and Brooks, 2007*). This assumption was also challenged by studies of the maturation of other HIV-1 Abs: for example, CH103, a non-VRC01-class CD4-binding site bNAb, exhibits changes in the relative orientation of its V$_H$ and V$_L$ domains during maturation (*Fera et al., 2014*), and the HIV-1 gp41-directed bNAb 4E10 exhibits increased flexibility in its combining site during maturation (*Finton et al., 2014*). Thus, the rare events that result in evolution of HIV-1 bNAbs can fall outside of typical Ab maturation pathways.

We speculate that 426c-based immunogens are able to bind germline precursors of VRC01-class bNAbs as a complete gp120 core because they can engage the unbound conformation of these Abs. This may allow the gp120-based immunogens to overcome a loss of binding affinity due to slow on-rates in the context of the already weak binding affinities of immature B cell receptors. The improved binding of the 426c-based gp120 immunogens to some germline VRC01-class Abs upon removal of the V1V2 and V3 loops from the 426c.TM1△V1-3 and 426c.TM4△V1-3 gp120s could result from the removal of steric occlusion by the variable loops (also not present in the eOD immunogens (*Jardine et al., 2013*; *2015*), and which may be flexible in the context of a gp120 or eOD), removal of glycans attached to these loops, or a combination of these factors. Thus, our results suggest implementation of a general strategy in which a germline-binding antigen is designed to be electrostatically compatible with the neutral or negatively charged antigen-binding surfaces of germline VRC01 Fabs and to fit in lock-and-key mode to the combining site of a germline Fab, with consideration paid to the slightly different angles of approach reported here for germline Fab binding to the gp120-based immunogens. The general principles established here for germline VRC01-class recognition of gp120 can be used to guide efforts to design and produce immunogens capable of eliciting broad and potent CD4bs Abs of the VRC01 class in uninfected people, facilitating the development of an efficacious vaccine to protect from HIV-1 infection.

## Materials and methods

### Protein expression and purification

NIH45-46$_{GL}$ and 3BNC60$_{GL}$ were constructed as described previously by using the VH1-2*02 germline V gene segment, the appropriate germline V$_L$ gene segment, and mature sequences for CDRH3 and CDRL3 (*Scharf et al., 2013*; *Hoot et al., 2013*; *McGuire et al., 2013*; *Dosenovic et al., 2015*). The Abs were expressed and purified as described (*Scharf et al., 2013*). Briefly, IgGs and 6xHis-tagged Fab fragments were produced by transient cotransfection of appropriate HC and LC plasmids into HEK293-6E cells followed by purification of the secreted proteins from cell supernatant using protein A (GE Healthcare; Pittsburg, PA) or Ni-NTA (GE Healthcare) affinity chromatography and Superdex 200 16/60 (GE Healthcare) size exclusion chromatography (SEC). gp120 proteins were expressed as cores with N/C termini and V1-V2 and V3 loop truncations as described for previous structural studies (*Zhou et al., 2010*) by transient transfection of suspension-adapted HEK293-S cells. gp120s were purified using Ni-NTA affinity chromatography and Superdex 200 16/60 SEC. Proteins were stored in 20 mM Tris, pH 8.0, and 150 mM sodium chloride (TBS buffer) supplemented with 0.02% (wt/vol) NaN$_3$.

## Crystallization

Crystals of 3BNC60$_{GL}$ Fab were obtained by combining 0.2 µL of a 15 mg/mL protein solution with 0.2 µL of 0.1 M bicine pH 9.1 and 10% (w/v) PEG 1,500 at 20°C and cryoprotected in mother liquor supplemented with 20% (v/v) glycerol. Crystals of 426c.TM4△V1-3 gp120 were obtained by combining 0.2 µL of a 10 mg/mL protein solution with 0.2 µL of 0.1 M sodium citrate tribasic dihydrate pH 5.0, 10% (w/v) PEG 6000 and 0.2 M sodium thiocyanate at 20°C and cryoprotected in mother liquor supplemented with 30% (v/v) 2-propanol. Complexes of 3BNC60$_{GL}$/426c.TM4△V1-3, NIH45-46$_{GL}$/426c.TM1△V1-3 and 3BNC55$_{MAT}$/426c.TM4△V1-3 were produced by incubating Fabs and gp120s at a 3:1 molar ratio at 4°C for 16 hrs, followed by Endoglycosidase H (New England BioLabs; Ipswich, MA) treatment and SEC purification. Fractions containing complexes were combined and concentrated as indicated below. Crystals of 3BNC60$_{GL}$/426c.TM4△V1-3 complex were obtained by combining 0.2 µL of a 20 mg/mL protein solution with 0.2 µL of 0.1 M imidazole pH 7.0, 8% (w/v) PEG10000, 10 mM calcium chloride dihydrate at 20°C and cryoprotected in mother liquor supplemented with 20% (w/v) ethylene glycol. Crystals of NIH45-46$_{GL}$/426c.TM1△V1-3 complex were obtained by combining 0.2 µL of a 20 mg/mL protein solution with 0.2 µL of 0.1 M sodium citrate pH 5.5, 22% (w/v) PEG 1000, 3% (w/v) xylitol at 20°C and cryoprotected in mother liquor supplemented with 20% (w/v) ethylene glycol. Crystals of 3BNC55$_{MAT}$/426c.TM4△V1-3 complex were obtained by combining 0.2 µL of a 15 mg/mL protein solution with 0.2 µL of 0.1 M sodium citrate pH 5.5, 18% (w/v) PEG 1,000, 3% (w/v) ethylene glycol at 20°C and cryoprotected in mother liquor supplemented with 20% (w/v) ethylene glycol. All crystals were flash cooled in liquid nitrogen.

## Data collection and structure determination

X-ray diffraction data were collected at the Stanford Synchrotron Radiation Lightsource beamline 12–2 outfitted with a Pilatus 6M pixel detector (Dectris; Baden-Dättwil, Switzerland). XDS was used to index, integrate and scale the data (*Kabsch, 2010*). The structures were refined using an iterative approach of refinement with Phenix (*Adams et al., 2010*) and manual model building in Coot (*Emsley and Cowtan, 2004*). Crystals of 3BNC60$_{GL}$ Fab (one molecule per asymmetric unit) diffracted to 1.9 Å, and the structure was solved by molecular replacement using 3BNC117$_{MAT}$ Fab (PDB code 4JPV) V$_H$V$_L$ with CDR loops removed and C$_H$1C$_L$ as search models. The final model (R$_{work}$ = 19.6%, R$_{free}$ = 21.0%) has 99%, 1%, and 0% of residues in the favored, allowed and disallowed regions, respectively, of the Ramachandran plot. Crystals of NIH45-46$_{GL}$/426c.TM1△V1-3 complex (one Fab-gp120 complex per asymmetric unit) diffracted to 3.4 Å, and the structure was solved by molecular replacement using 93TH057 gp120 core (PDB code 4JDT) and NIH45-46$_{GL}$ Fab V$_H$V$_L$ with CDR loops removed and C$_H$1C$_L$ (from PDB code 4JDV) as the search models. The final model (R$_{work}$ = 27.9%, R$_{free}$ = 28.6%) has 95.7%, 4.3% and 0% of residues in the favored, allowed, and disallowed regions, respectively, of the Ramachandran plot. Some parts of the C$_H$1C$_L$ domain were not well ordered in the electron density, probably because this domain did not make crystal packing contacts. To account for regions of disorder, unresolved C$_H$1C$_L$ residues were assigned occupancies of 0. Crystals of 426c.TM4△V1-3 gp120 diffracted to 2.0 Å, contained two molecules in the asymmetric unit and the structure was solved by molecular replacement using 426c.TM1△V1-3 as the search model. The final model (R$_{work}$ = 20.7%, R$_{free}$ = 23.2%) has 97.7%, 2.3% and 0% of residues in the favored, allowed, and disallowed regions, respectively, of the Ramachandran plot. Crystals of 3BNC60$_{GL}$/426c.TM4△V1-3 complex diffracted to 3.1 Å, contained two Fab-gp120 complexes and two unbound gp120 molcules in the asymmetric unit and the structure was solved by molecular replacement using 426c.TM4△V1-3 gp120 core and 3BNC60$_{GL}$ Fab V$_H$V$_L$ and C$_H$1C$_L$ with CDR loops removed as the search models. The final model (R$_{work}$ = 20.3%, R$_{free}$ = 26.7%) has 97%, 2.9% and 0.1% of residues in the favored, allowed and disallowed regions, respectively, of the Ramachandran plot. Crystals of 3BNC55$_{MAT}$/ 426c.TM4△V1-3 complex diffracted to 3.0 Å, contained two Fab-gp120 complexes in the asymmetric unit and the structure was solved by molecular replacement using 426c.TM4△V1-3 gp120 core and 3BNC117$_{MAT}$ Fab (from PDB code 4JPV) V$_H$V$_L$ and C$_H$1C$_L$ with CDR loops removed as the search models. The final model (R$_{work}$ = 24.0%, R$_{free}$ = 27.9%) has 98%, 1.7%, and 0.3% of residues in the favored, allowed, and disallowed regions, respectively, of the Ramachandran plot.

Buried surface areas were calculated using PDBePISA (*Krissinel and Henrick, 2007*) and a 1.4 Å probe. Hydrogen bonds were assigned tentatively due to the low resolution of the complex

structures using the following criteria: a distance of <3.5 Å, and an A-D-H angle of >90°. Structures were superimposed and molecular representations were generated with PyMOL (*Schrödinger LLC, 2011*) and UCSF Chimera (*Pettersen et al., 2004*). Rmsd calculations following pairwise Cα alignments were done in PyMOL without outlier rejection.

### SPR experiments

All SPR measurements were performed on a Biacore T200 (GE Healthcare) at 20°C using HBS-EP+ (GE Healthcare) as the running buffer. A CM5 chip (GE Healthcare) containing 3000 resonance units (RUs) of primary amine-coupled protein A (Pierce; Waltham, MA) was used to capture HIV-1 IgGs (3BNC60$_{GL}$, NIH45-46$_{GL}$, VRC01$_{GL}$, 3BNC60$_{MAT}$, NIH45-46$_{MAT}$) and a non-gp120-binding control IgG (mG053) by injecting 0.25 µg/mL or 0.1 µg/mL solutions of germline or mature IgG, respectively. Remaining protein A binding sites were blocked by injecting 1 µM Fc. gp120 cores (426c. TM1△V1-3, 426c.TM4△V1-3, 93TH057) were injected over the flow cells at increasing concentrations (top concentration of 16 µM for germline Abs, 200 nM for mature Abs) at a flow rate of 50 µL/min for 240 s and allowed to dissociate for 500 sec. Regeneration of flow cells was achieved by injecting one pulse each of 10 mM glycine pH 2 and 1 M guanidine-HCl at a flow rate of 90 µL/min. Kinetic analyses were used after subtraction of reference curves to derive on/off rates ($k_a$/$k_d$) and binding constants ($K_D$s) with a 1:1 binding model with or without bulk refractive index change (RI) correction as appropriate (Biacore T200 Evaluation software).

### Antibody approach angle comparisons

The VRC01$_{MAT}$/93TH057 gp120 complex (PDB code 3NGB) was used as the reference structure for comparisons of angles of approach of Fab recognition of gp120s (*Figure 7B*). The center of mass of the VRC01 V$_H$ domain was placed at the origin and its principal axes of inertia were aligned with the Cartesian axes using AMORE from the CCP4 program suite (*CCP4, 1994*). The 3NGB complex was then aligned with the centered V$_H$ domain. For comparisons with other complexes, each Fab/gp120 complex was aligned with the 3NGB gp120 chain using LSQMAN (*Kleywegt, 1996*). The transformation matrix between the aligned Fab/gp120 V$_H$ domain and the VRC01 V$_H$ domain was then calculated by LSQMAN.

## Acknowledgements

We thank the Caltech Protein Expression Center for producing antibody and gp120 proteins, generation of suspension-adapted HEK293-S cells and use of the Biacore T200. Operations at the Stanford Synchrotron Radiation Lightsource are supported by the US Department of Energy and the National Institutes of Health. MCN is a Howard Hughes Medical Institute investigator. This research was supported by the National Institute Of Allergy And Infectious Diseases of the National Institutes of Health Grant HIVRAD P01 AI100148 (PJB and MCN); (the content is solely the responsibility of the authors and does not necessarily represent the official views of the National Institutes of Health), Collaboration for AIDS Vaccine Discovery grants from The Bill and Melinda Gates Foundation (grant IDs 1040753 to PJB and 1124068 to MCN), the American Cancer Society (Grant PF-13-076-01-MPC to LS), and the California HIV/AIDS Research Program (CHRP grant F12-CT-214 to SAS).

## Additional information

### Competing interests

MCN: Reviewing editor, *eLife*. The other authors declare that no competing interests exist.

### Funding

| Funder | Grant reference number | Author |
| --- | --- | --- |
| National Institutes of Health | AI100148 | Michel C Nussenzweig Pamela J Bjorkman |
| Bill and Melinda Gates Foundation | 1040753 | Pamela J Bjorkman |
| American Cancer Society | PF-13-076-01-MPC | Louise Scharf |

| California HIV/AIDS Research Program | F12-CT-214 | Stuart A Sievers |
| Bill and Melinda Gates Foundation | 1124068 | Michel C Nussenzweig |

The content is solely the responsibility of the authors and does not necessarily represent the official views of the National Institutes of Health, the Bill and Melinda Gates Foundation, ACS or CHRP.

### Author contributions

LSc, Conception and design, Acquisition of data, Analysis and interpretation of data, Drafting or revising the article; APW, Analysis and interpretation of data, Drafting or revising the article; SAS, CC, SJ, Acquisition of data, Analysis and interpretation of data; HG, Acquisition of data, Contributed unpublished essential data or reagents; MDG, JFS, Contributed unpublished essential data or reagents; ATM, MCN, LSt, Drafting or revising the article, Contributed unpublished essential data or reagents; PJB, Conception and design, Analysis and interpretation of data, Drafting or revising the article

## Additional files

### Supplementary files

• Supplementary file 1. Interfaces in NIH45-46$_{GL}$-426c.TM1ΔV1-3 and 3BNC60$_{GL}$-426c.TM4ΔV1-3 Complexes.

### Major datasets

The following datasets were generated:

| Author(s) | Year | Dataset title | Dataset URL | Database, license, and accessibility information |
|---|---|---|---|---|
| Scharf L, Sievers SA, Jiang S, Bjorkman PJ | 2015 | Crystal structure of germ-line precursor of 3BNC60 Fab | http://www.rcsb.org/pdb/search/structid-Search.do?structureId=5F7E | Publicly available at the RCSB Protein Data Bank (accession no. 5F7E) |
| Scharf L, Bjorkman PJ | 2015 | Crystal structure of 426c.TM4deltaV1-3 gp120 | http://www.rcsb.org/pdb/search/structid-Search.do?structureId=5FA2 | Publicly available at the RCSB Protein Data Bank (accession no. 5FA2) |
| Scharf L, Bjorkman PJ | 2015 | Crystal structure of 3BNC60 Fab germline precursor in complex with 426c.TM4deltaV1-3 gp120 | http://www.rcsb.org/pdb/search/structid-Search.do?structureId=5FEC | Publicly available at the RCSB Protein Data Bank (accession no. 5FEC) |
| Scharf L, Bjorkman PJ | 2015 | Crystal structure of NIH45-46 Fab germline precursor in complex with 426c.TM1deltaV1-3 gp120 | http://www.rcsb.org/pdb/search/structid-Search.do?structureId=5IGX | Publicly available at the RCSB Protein Data Bank (accession no. 5IGX) |
| Scharf L, Chen C, Bjorkman PJ | 2015 | Crystal structure of 3BNC55 Fab in complex with 426c.TM4deltaV1-3 gp120 | http://www.rcsb.org/pdb/search/structid-Search.do?structureId=5I9Q | Publicly available at the RCSB Protein Data Bank (accession no. 5I9Q) |

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
