## [Decision Letter]

Thank you for submitting your work entitled "Structural basis for germline antibody recognition of gp120-based HIV-1 immunogens" for consideration by *eLife*. Your article has been favorably evaluated by three reviewers, one of whom is a member of our Board of Reviewing Editors, and the evaluation was overseen by Tadatsugu Taniguchi as Senior editor.

The reviewers have discussed the reviews with one another and the Reviewing Editor has drafted this decision to help you prepare a revised submission. Please find below the general and specific comments.

The induction of bnAbs against HIV by vaccination will undoubtedly require the activation of their germline precursors. The design of such immunogens is a major problem. In this work, the interactions made between germline VRC01-class Abs and gp120 constructs were studied. Five new crystal structures were solved: one unliganded germline VRC01-class Ab, one unliganded gp120 that has been modified for germline Ab binding, two complex structures of GL VRC01-class Abs bound to the engineered gp120 core and one complex structure of a VRC01-class bnAb bound to the engineered gp120 core. An in depth comparison between these five structures as well as previously solved structures is carried out to evaluate conformational changes in the GL Abs and bnAbs, gp120/Ab contacts, buried surface area, angles of approach, glycan interactions, and electrostatics.

While, as noted by the authors, much of the analysis of angle of approach, BSA, conserved contacts and preformed conformation of the Abs can already be found in West/Bjorkman (PNAS, 2012), Diskin/Bjorkman (J Exp Med, 2013), Jardine/Schief (Science, 2013) and Scharf/Diskin (PNAS, 2013), since structures of a complex between a fully germline VRCO1-class Ab and a complex gp120 core were not available, the addition of the five crystal structures, especially the 426c bound structures, is valuable to the field. Moreover, many of the findings derived from the structures are significant. However, the following points need to be addressed in a revised manuscript.

1) Comparison of "binding mode" as reported by translation distance and rotation angle: this discussion is interesting but incomplete and requires further clarification.

The wire diagrams in Figure 7 imply that the portions of the antibodies proximal to gp120 are much better aligned than the distal portions. This begs the question: Might the distance/angle differences in Figure 7 be due to protein flexibility in the Fab domains accommodating differences in crystal packing? This point should be carefully examined and discussed to support the relevance of the measurements to vaccine design or therapeutic antibody design.

The true target of these antibodies is a native trimer that may restrict the translation distance and angle of approach more severely than the core-gp120s in the crystal structures analyzed here. Because these antibodies are broadly neutralizing, we already know that they can bind native trimers of diverse strains. Are the authors confident that the variations in distance and angle reported here would be reproduced if crystal structures were solved with the same antibodies in complex with native-like trimers? A key uncertainty in this regard is caused by the bridging sheet, an important aspect of the VRC01-class interaction with gp120 that has been shown to be in very different conformations in core-gp120 structures (complexed with CD4 or VRC01-class bnAbs) as compared to native-like trimer structures (Lyumkis et al. Science 2013, Figure 5). Further uncertainty comes from the V2 loop that is known to restrict access to the CD4 binding site on the trimer, yet is absent from the structures analyzed here. Finally, the many glycans around the CD4 binding site on the trimer may restrict the translation distance and measured angle. The authors should attempt to use the published structures of the BG505 SOSIP native-like trimer and other native-like trimers to assess the relevance of their distance/angle measurements on core-gp120 to the interaction of VRC01-class bnAbs with native-like trimers.

The authors have computed translation distance and rotation angle only for a subset of the VRC01-class bnAbs for which high resolution structures are available. This incomplete analysis means that the distance/angle distribution for VRC01-class bnAbs interacting with core-gp120 might be broader (or narrower) than currently represented in Figure 7. The incomplete analysis leaves open the possibility that the distance/angle values for the TM1 and TM4 structures may not be as divergent from the previously known structures as implied by Figure 7. The authors should include analysis of several (if not all) currently available structures of VRC01-class bnAbs in complex with core-gp120.

The authors should comment on the fact that the eOD is a more minimal antigen than 426c yet it appears to better preserve the distance/angle of the VRC01-class interaction (Figure 7). Can the authors discern the structural reason why 426c supports a more unusual distance/angle interaction mode even though it contains the inner domain and bridging sheet that are absent from eOD? This type of analysis could help guide further immunogen design.

The authors have computed translation distance and rotation angle for one antibody (3BNC117MAT) crystallized with three different gp120 molecules, but they should provide some discussion of the variation if any in the distance/angle values among these three structures. This would provide a useful reference for the effects of gp120 strain variation and crystal packing on the distance/angle values that are being measured.

2) The discussion of how the electrostatics of the VRC01-class Abs' interaction is changed by affinity maturation is very interesting. The reason offered by the authors is that this helps binding to the glycans. But, the arguments pertinent to this issue in the Discussion – are a bit confusing. At the start of the fifth paragraph it is stated that interactions with glycans are facilitated by the change in electrostatic potential. Later in that paragraph it is stated that this helps avoiding interactions with the glycan. Some rewording may be appropriate. Perhaps, the authors mean to say that somatic hypermutation on the LC has two objectives-the main one being to avoid clashes with the N276 glycan that will block binding, and secondly to make additional contacts to the glycan?

Even though this finding is of interest, the discussion of this being related to interactions with complex glycans with negatively charged sialic acids needs to be tempered. The degree to which the N276 glycan or other glycans near the VRC01-class epitope are complex type rather than high mannose is not well understood. The existing data on glycosylation profiles of HIV Env are incomplete as they were largely measured on gp120 molecules rather than native-like trimers.

3) The discussion of lock-and-key versus induced-fit binding mechanisms is simplistic because in reality there is likely a continuum of binding mechanisms rather than only two ways to bind. Could this discussion be nuanced?

4) The authors paint a picture that VRC01-class development involves "lock-and-key" germline Fab interaction with increasing flexibility during maturation. However, perhaps the Discussion should be modified to account for the following:

It is interesting that the 426c gp120 interaction with germline Fabs appears to involve little conformational change in the Fab. However, the authors seem to be making a leap when they attempt to contrast this finding with their previous observation that the CDRH3 of NIH45-46_CHIM_ was disordered in the interaction with an "unmodified" gp120 is (Discussion, eighth paragraph). Could that difference not also be due to different crystal packing? Alternatively, could this be due to interaction with a different strain of gp120? Or due to the fact that one antibody is "germline" while the other is a chimera of "germline"-H with mature-L? It seems difficult to draw a strong conclusion from the contrast given. This is compounded by the fact that the CDRH3 makes little contact in most of the VRC01-class/core-gp120 crystal structures, hence whether the CDRH3 is ordered or not may not be highly important to the interaction.

To support the idea that VRC01-class maturation involves increasing flexibility, the authors then give the example (Discussion, eighth paragraph) of the Ala61_HC_Pro mutation in the 3BNC60/3BNC117 members of this class. However, the generality of this example is unclear. Do other members of the class exhibit similar features?

---

## [Author Response]

*[T]he following points need to be addressed in a revised manuscript. 1) Comparison of "binding mode" as reported by translation distance and rotation angle: this discussion is interesting but incomplete and requires further clarification.*

*The wire diagrams in Figure 7 imply that the portions of the antibodies proximal to gp120 are much better aligned than the distal portions.*

The different Ab-gp120 complexes were compared by aligning their gp120 components. The closer alignment of the portions of the antibodies proximal to gp120 arises by necessity because differences would be expected to radiate out from the common point of alignment (the gp120).

*This begs the question: Might the distance/angle differences in Figure 7 be due to protein flexibility in the Fab domains accommodating differences in crystal packing? This point should be carefully examined and discussed to support the relevance of the measurements to vaccine design or therapeutic antibody design.*

It is extremely unlikely that the interaction between the combining site of an antibody and its target antigen could be altered by crystal packing because these interactions result from intermolecular protein-protein contacts that span large surface areas, whereas crystal contacts bury relatively little surface area compared with interactions between partners in a protein-protein interaction (e.g., see J Janin, 1997, “Specific versus non-specific contacts in protein crystals” NSMB 4, 973-974). If the interaction between an antibody combining site and its antigen can be altered by crystal packing, then much of what has been learned from protein crystallography about antibody-antigen interactions (and protein-protein interactions in general) would be suspect.

*The true target of these antibodies is a native trimer that may restrict the translation distance and angle of approach more severely than the core-gp120s in the crystal structures analyzed here. Because these antibodies are broadly neutralizing, we already know that they can bind native trimers of diverse strains. Are the authors confident that the variations in distance and angle reported here would be reproduced if crystal structures were solved with the same antibodies in complex with native-like trimers?*

It’s not actually clear what the antigen target of germline antibodies is – it could be that shed gp120s bind to and activate germline BCRs. In any case, the 426c-based immunogens were designed to bind as gp120s, not as trimers, to germline antibodies. Indeed, the germline antibodies that were investigated in our study do not bind to native-like trimers or even to unaltered gp120s in general. Since currently available BG505 SOSIP trimer structures have no features that would prohibit the range of binding orientations we observed (see Figure 7 in revised manuscript), we would expect these variations to be reproduced in structures of corresponding BG505 SOSIP-Fab complexes.

The only published structures relevant to the reviewer’s question of whether the orientation observed for a mature VRC01-class Fab/gp120 complex is predictive of the orientation of the same Fab bound to a native-like trimer are the cryo-EM structure of PGV04 (also known as VRC-PG04) Fab bound to the BG505 SOSIP trimer (Lyumkis et al., 2013) and the structures of this Fab with gp120s (PDB codes 3SE9, 4I3S, and 4I3R). The PGV04/trimer structure yields a V_H_ rotation and translation of 6.1° and 2.5 Å compared to the VRC01 reference, only modestly different from the values calculated for the PGV04/gp120 structure 3SE9 (5.6° and 0.8 Å).

*A key uncertainty in this regard is caused by the bridging sheet, an important aspect of the VRC01-class interaction with gp120 that has been shown to be in very different conformations in core-gp120 structures (complexed with CD4 or VRC01-class bnAbs) as compared to native-like trimer structures (Lyumkis et al.Science 2013, Figure 5). Further uncertainty comes from the V2 loop that is known to restrict access to the CD4 binding site on the trimer, yet is absent from the structures analyzed here. Finally, the many glycans around the CD4 binding site on the trimer may restrict the translation distance and measured angle. The authors should attempt to use the published structures of the BG505 SOSIP native-like trimer and other native-like trimers to assess the relevance of their distance/angle measurements on core-gp120 to the interaction of VRC01-class bnAbs with native-like trimers.*

Regarding the bridging sheet, while its conformation differs between Env trimers and gp120 monomers, and some CD4bs Abs make close contacts with this region, for the VRC01-like bNAbs, the bridging sheet in the trimer is mostly out of range for direct contacts. This is indicated in the Lyumkis et al., 2013, Figure 5 legend, which refers to “minimal clashes with PGV04” [a VRC01-like bNAb], as compared with predicted clashes with non-VRC01-class CD4bs bNAbs. Alignment of the 3BNC60_GL_ Fab-426c gp120 structure with the Env trimer of PDB entry 5CEZ indicates the closest distance between the Fab and the bridging sheet strands that have different topology in the trimer (beta2-beta3) is about 9 Å (Fab HC Ser75 to Env residue 198).

The reviewers point out that the V2 loop and glycans restrict access to the CD4 binding site on the trimer. However, the V1V2 and V3 loops, as well as some of the N-glycans around the CD4 binding site, were removed from the 426c-based gp120 immunogens in order to permit them to engage germline versions of VRC01-class bNAb B cell receptors. The presence of these features on native-like Env trimers likely explains why they are not recognized by germline versions of CD4-binding site bNAbs (e.g., McGuire et al., 2016, Nature Comm) shows that 3BNC60_GL_ does not bind the 426c gp120 when the variable loops are present).

Nevertheless, we can model a germline Fab onto a BG505 SOSIP structure to see if the altered orientation sterically clashes with adjacent subunits in the trimer. This modeling exercise suggests that the different orientation of the 3BNC60_GL_ Fab with respect to gp120 moves it away from the adjacent gp120 subunit, suggesting this orientation would be sterically compatible when binding to Env trimer (shown in Figure 7 new figure in the revised manuscript). In this model the closest Cα–Cα distance between the 3BNC60_GL_ Fab and the adjacent gp120 is about 6.5 Å.

Modeling clashes with glycans present in any trimer-Fab (or gp120-Fab) molecular structure would offer limited insight into the actual steric hindrance that antibodies have to accommodate. With the exception of the N-acetyl glucosamine residue attached directly to the Asn of an *N*-linked glycosylation site, glycans have no rigid orientation or position. Furthermore, a glycan conformation observed in a high resolution structure is due to contacts it makes with other molecules as this conformation is only one of many conformations accessible to that glycan (e.g., the Asn276_gp120_ glycan is ordered in the structure of a VRC01-class Fab/gp120 complex (45-46m2/93TH057; Diskin et al., 2013) and in structures of 8ANC195 (not a CD4bs bNAb) bound to gp120 or to BG505 SOSIP (Scharf et al., 2014; 2015), but it occupies different positions when Env is bound to the CD4bs bNAb versus when it is bound to the 8ANC195 Fab).

*The authors have computed translation distance and rotation angle only for a subset of the VRC01-class bnAbs for which high resolution structures are available. This incomplete analysis means that the distance/angle distribution for VRC01-class bnAbs interacting with core-gp120 might be broader (or narrower) than currently represented in Figure 7. The incomplete analysis leaves open the possibility that the distance/angle values for the TM1 and TM4 structures may not be as divergent from the previously known structures as implied by Figure 7. The authors should include analysis of several (if not all) currently available structures of VRC01-class bnAbs in complex with core-gp120.*

This point is addressed in our explanation of the question below.

*The authors should comment on the fact that the eOD is a more minimal antigen than 426c yet it appears to better preserve the distance/angle of the VRC01-class interaction (Figure 7). Can the authors discern the structural reason why 426c supports a more unusual distance/angle interaction mode even though it contains the inner domain and bridging sheet that are absent from eOD? This type of analysis could help guide further immunogen design.*

The original Figure 7 included VRC01-class bNAbs from 8 of the 9 published independent clonal families of this class of bNAb. To provide a more comprehensive analysis, we have now included all cases with published structures, as well as including alignment results for all bNAb/Env complexes within the asymmetric unit of each crystal structure. The new version of Figure 7 includes 25 mature VRC01-class antibody alignments, as compared with 9 in the original version.

Unfortunately it is difficult to discern a structural reason for the differences in orientation in the eOD and 426c.TM△V1-3 complexes due to the extensive interfaces and large numbers of contacts.

*The authors have computed translation distance and rotation angle for one antibody (3BNC117MAT) crystallized with three different gp120 molecules, but they should provide some discussion of the variation if any in the distance/angle values among these three structures. This would provide a useful reference for the effects of gp120 strain variation and crystal packing on the distance/angle values that are being measured.*

The more comprehensive analysis now presented in Figure 7 (and detailed in [Supplementary-material SD1-data]) shows the distance/angles values are consistent among individual complexes in the asymmetric unit (which are subject to different crystal packing interactions). The analysis also shows consistency among complexes crystallized with different gp120s and between mutants/clonal variants.

*2) The discussion of how the electrostatics of the VRC01-class Abs' interaction is changed by affinity maturation is very interesting. The reason offered by the authors is that this helps binding to the glycans. But, the arguments pertinent to this issue in the Discussion are a bit confusing. At the start of the fifth paragraph it is stated that interactions with glycans are facilitated by the change in electrostatic potential. Later in that paragraph it is stated that this helps avoiding interactions with the glycan. Some rewording may be appropriate. Perhaps, the authors mean to say that somatic hypermutation on the LC has two objectives-the main one being to avoid clashes with the N276 glycan that will block binding, and secondly to make additional contacts to the glycan?*

Thank you for pointing out these points of confusion in the original manuscript. The revised paper states that the increased electropositivity could be part of a multi-pronged approach of the humoral immune response against the CD4 binding site of HIV-1 for accommodating and/or for avoiding the complex *N*-linked Env glycan attached to Asn276_gp120_.

*Even though this finding is of interest, the discussion of this being related to interactions with complex glycans with negatively charged sialic acids needs to be tempered. The degree to which the N276 glycan or other glycans near the VRC01-class epitope are complex type rather than high mannose is not well understood. The existing data on glycosylation profiles of HIV Env are incomplete as they were largely measured on gp120 molecules rather than native-like trimers.*

We agree and have revised the discussion of the potential interactions with negatively charged sialic acids. Regarding glycosylation profiles, we added a citation to a recent manuscript from Max Crispin’s laboratory (Behrens et al., 2016, in press) in which they have determined the site-specific glycan profile of BG505 SOSIP.664, a native-like trimer.

*3) The discussion of lock-and-key versus induced-fit binding mechanisms is simplistic because in reality there is likely a continuum of binding mechanisms rather than only two ways to bind. Could this discussion be nuanced?* The reviewers make an important point. We have revised the discussion of these antibody binding mechanisms to make the points in a more nuanced manner.

*4) The authors paint a picture that VRC01-class development involves "lock-and-key" germline Fab interaction with increasing flexibility during maturation. However, perhaps the Discussion should be modified to account for the following:*

*It is interesting that the 426c gp120 interaction with germline Fabs appears to involve little conformational change in the Fab. However, the authors seem to be making a leap when they attempt to contrast this finding with their previous observation that the CDRH3 of NIH45-46_CHIM_ was disordered in the interaction with an "unmodified" gp120 is (Discussion, eighth paragraph). Could that difference not also be due to different crystal packing?*

Neither the CDRH3 loop of the NIH45-46_CHIM_/93TH057 gp120 complex nor the CDRH3 loop of the NIH45-46_GL_/426c.TM1△V1-3 gp120 complex structure are involved in crystal packing interactions (see Figure 9).

Author response image 1.CDR3_HC_ positions in NIH45-46_CHIM_/93TH057 gp120 (left) and NIH45-46_GL_/426c.TM1△V1-3 gp120 (right) complexes relative to symmetry mates (different views shown for clarity).One complex of gp120 (grey) with antibody V_H_ (pink) and V_L_ (light pink) is shown with the nearest two symmetry mates (blue and green). The disordered region of NIH45-46_CHIM_ CDRH3 is indicated by a dashed line. The CDRH3 loop is not engaged in crystal packing interactions in either complex.**DOI:**
http://dx.doi.org/10.7554/eLife.13783.016

*Alternatively, could this be due to interaction with a different strain of gp120? Or due to the fact that one antibody is "germline" while the other is a chimera of "germline"-H with mature-L?*

Either of these reasons could account for the difference. Since a germline HC/germline LC NIH45-46 Fab did not bind to 93TH057 gp120, while the germline HC/mature LC chimeric Fab did, it seems likely that this difference is due to the inclusion of the mature LC.

*It seems difficult to draw a strong conclusion from the contrast given. This is compounded by the fact that the CDRH3 makes little contact in most of the VRC01-class/core-gp120 crystal structures, hence whether the CDRH3 is ordered or not may not be highly important to the interaction.*

Please see Table S4 from Diskin et al., 2011, Science in which we compared the surface areas buried on different structural elements of NIH45-46 and VRC01. The surface areas buried on CDR3 represent 28% (NIH45-46) and 13% (VRC01) of the total surface area buried on the Fab by gp120. Thus although VRC01-class bNAbs do not fall in the CDRH3-dominated class of CD4-binding site antibodies (as defined by Zhou et al., 2015, Cell), the CDRH3 loops make important contacts to gp120 that are critical for their interactions.

*To support the idea that VRC01-class maturation involves increasing flexibility, the authors then give the example (Discussion, eighth paragraph) of the Ala61_HC_Pro mutation in the 3BNC60/3BNC117 members of this class. However, the generality of this example is unclear. Do other members of the class exhibit similar features?*

The 3BNC clonal family (3BNC60/3BNC117) is the only set of VRC01-class antibodies that includes the Ala61Pro somatic hypermutation. This substitution nonetheless represents an example of increased flexibility during antibody maturation.